# The Impact of Geometric Complexity on Neural Collapse in Transfer Learning

**Michael Munn**[*]
Google Research
munn@google.com

**Benoit Dherin**[*]
Google Research
dherin@google.com

**Javier Gonzalvo**
Google Research
xavigonzalvo@google.com

## Abstract

Many of the recent remarkable advances in computer vision and language models can be attributed to the success of transfer learning via the pre-training of large foundation models. However, a theoretical framework which explains this empirical success is incomplete and remains an active area of research. Flatness of the loss surface and neural collapse have recently emerged as useful pre-training metrics which shed light on the implicit biases underlying pre-training. In this paper, we explore the geometric complexity of a model's learned representations as a fundamental mechanism that relates these two concepts. We show through experiments and theory that mechanisms which affect the geometric complexity of the pre-trained network also influence the neural collapse. Furthermore, we show how this effect of the geometric complexity generalizes to the neural collapse of new classes as well, thus encouraging better performance on downstream tasks, particularly in the few-shot setting.

## 1 Introduction

Many of the recent remarkable advances in modern machine learning owe their success in part to transfer learning [4, 6, 9, 15, 28, 49, 50, 66]. While there are different approaches to this technique, the standard one involves two stages. In a first stage, called pre-training, ones trains a deep neural network on a general, large-scale dataset in the form of a supervised or unsupervised source task; e.g., ImageNet or CIFAR-100 [14, 33] for image models or the Common Crawl, C4 or LM1B datasets [8, 11, 49] for language models. In the second stage, one then leverages portions of the pre-trained network to use as features map or embeddings that can then be adapted, or fine-tuned, on a more specific target task. Often these target tasks are unknown at the time of pre-training and labeled data may be very scarce, such as in the context of few-shot learning [34, 51, 60]. However, despite these limitations, this approach often results in a fine-tuned model that achieves quite impressive performance substantially better than training on the target task alone and requiring less computational resources [20]. Despite this empirical success, a comprehensive theoretical understanding of the mechanisms underlying this effectiveness of transfer learning is not fully understood and remains an active area of research [63].

One interesting line of research suggests that the effectiveness of transfer learning is due to the implicit biases encoded in pre-trained models [22, 36, 48]. These implicit biases effectively constrain the hypothesis space, guiding the model towards solutions that have a preference for smoother functions [12, 17, 38, 40, 53], simpler geometry in the loss surface [3, 13, 18, 55, 58] or reduced complexity of the internal learned representations [23, 32]. In the same way that these implicit biases have been used to help explain the success of deep learning, recent work has shown that the notion of neural collapse [22, 35, 62] and flatness of the loss surface [36] can also inform the mechanisms behind

---

[*]Equal contribution.

38th Conference on Neural Information Processing Systems (NeurIPS 2024).

transfer learning. This suggests that these preferences during pre-training also act as a form of prior knowledge which is highly transferable to downstream tasks, even with limited task-specific data.

In this paper, we present a novel perspective that further sheds light on these mechanisms and implicit biases hidden within transfer learning. Our approach analyzes the geometric complexity [16, 17] of the internal representations learned by the deep neural networks during pre-training and provides a complementary theoretical framework which unifies previous work examining the role of neural collapse and loss surface flatness in transfer learning [22, 36]. In particular, we show through experiments and theory that the geometric complexity of the pre-trained network directly controls the neural collapse of the pre-trained model and thus its efficacy in transfer learning, particularly in the few-shot setting. We argue that geometric complexity (similarly to flatness of the loss surface and neural collapse) can be used as hidden progress metrics, cf. [2], for transfer learning, serving as an informative proxy toward the transfer power of a pre-trained network.

Our primary contributions are the following:

- We uncover relationships between learning-path flatness, neural network geometric-complexity, and embedding neural-collapse providing a framework to understand how these implicit biases interact (Section 4).
- We show through theory that the geometric complexity (GC) of a neural network controls neural collapse and verify this empirically by showing how mechanisms which regularize the GC in turn put pressure on the neural collapse (Section 4.1 and Fig. 1).
- We demonstrate both theoretically and empirically that pre-trained networks with lower GC promote lower neural collapse on new unseen target classes, and thus enable improved transfer accuracy during fine-tuning (Section 5 and Fig. 4).
- We prove a new generalization bound in terms of geometric complexity (Section 4.3).
- We show that the empirical GC can be accurately and efficiently estimated on a small number of samples, input coordinates, and output features making it computationally tractable compared to other progress metrics in machine learning (Section 4.2 and Fig. 2).

**Notation.**    Throughout, we denote by $\| \cdot \|$ the L2 Euclidean norm and by $\| \cdot \|_F$ the Frobenius norm.

## 2    Background and Related Work

The implicit biases introduced by an optimization algorithm play a crucial role in deep learning and in the generalization ability of the learned models [43, 44, 55, 64]. They help ensure that the model not only finds a solution with low error but also one with low complexity which generalizes well [26, 67]. Uncovering the mechanisms of these implicit regularizers is crucial for understanding how the model learns, both as a means to improve generalization and as valuable leverage for designing more efficient algorithms and decreasing costly experiment iteration cycles.

Here we focus on three seemingly different themes behind implicit regularization in deep learning and explain how they are related: the loss surface slope measuring the flatness of the learning path [3, 29, 45], the geometric complexity of the learned model function measuring its variability with respect to a dataset [16, 42], and the neural collapse [32, 46] measuring how a neural network efficiently clusters its learned representations of the input class examples in embedding space.

**Flatness in parameter space.**    In parameter space, the learning dynamics is fully characterized by the discrete optimization path $\theta_t$ where $t \in \mathbb{N}$. At each step, one can compute the flatness of the learning path as the slope of the tangent space at $\theta_t$ to the loss surface $\mathcal{L} = \{(\theta, L(\theta)) : \theta \in \mathbb{R}^n\}$. As shown in [3, Appendix A.2], this slope coincides with the loss-gradient square norm: $\text{slope}(\theta_t) = \|\nabla L(\theta_t)\|^2$. This quantity has been used to bound a generalization gap in [25] and as a beneficial explicit regularizer in [3, 24, 58] indicating that learning curves with lower slope values tend to also have better test performance.

Moreover, many standard optimizers and common training heuristics have been shown to put an implicit pressure on the learning-path flatness, making it an implicit bias of the training procedure [3, 5, 10, 13, 17, 25, 39, 58]. Related to the learning curve slope is the optima sharpness

$$\text{sharpness}(\theta_*) = \frac{1}{n} \text{trace} \, H(\theta_*)$$

where $\theta_*$ is a global minima toward which the learning dynamics converges to $\theta_t \to \theta_*$ and $H$ is the Hessian of the loss. Since the sharpness of an optima corresponds to the mean curvature of the loss surface at that point, flat minima (i.e., minima with low curvature in all directions) can be reached only through learning paths with shallower slopes.

**Neural collapse in embedding space.** Neural collapse [22, 27, 46] refers to a phenomenon observed in deep learning where the embedding network (i.e., the subnetwork $f$ before the logit layer $g$ in a neural network $h(x) = \mathrm{softmax}(g(f(x)))$ collapses the input points around their respective class means. Furthermore, these class means form a simplex creating an equiangular tight frame (ETF) centered around the global mean with roughly equal distance between its vertices. Intuitively, such a phenomenon is beneficial to generalization since the embeddings of different classes are optimally separated, making the job of the classifier head $\mathrm{softmax}(g(z))$ easier. To measure neural collapse, the authors in [22] introduce the *class-distance normalized variance* (CDNV) as

$$V_f(Q_i, Q_j) := \frac{\mathrm{Var}_f(Q_i) + \mathrm{Var}_f(Q_j)}{2\|\mu_f(Q_i) - \mu_f(Q_j)\|^2}, \tag{1}$$

where $Q_r = q_r(x)dx$ is the input distribution for classes $r \in \{i, j\}$ and where

$$\mu_f(Q_r) = \mathbb{E}_{x \sim Q_r}[f(x)] \quad \text{and} \quad \mathrm{Var}_f(Q_r) = \int \|\mu_f(Q_r) - f(x)\|^2 q_r(x)dx. \tag{2}$$

Neural collapse between two classes happens when their class variance decreases while the distance between their class means increases, causing lower values of their CDNV. Following [22], given a well-balanced input distribution $Q = \frac{1}{k}(Q_1 + \cdots + Q_k)$ with $k$ classes ($Q_i$ being the input distribution for class $i$), *neural collapse* (NC) during training is characterized by the following limit as the number of training steps $t$ goes to infinity:

$$\lim_{t \to \infty} \mathrm{NC}(f_t, Q) = 0 \quad \text{where} \quad \mathrm{NC}(f, Q) := \mathrm{Avg}_{i \neq j}\left(V_f(Q_i, Q_j)\right), \tag{3}$$

and where $f_t$ is the learned embedding network at step $t$. Thus, more neural collapse (i.e., more clustering around better separated class-means) happens with lower NC values. Proposition 5 from [22] proves that $\mathrm{NC}(f, Q)$ bounds a generalization gap and lower values of $\mathrm{NC}(Q, f)$ are correlated with better test performance of the neural network $f$, and [37] shows that explicitly regularizing for neural collapse is beneficial.

**Geometric complexity in function space:** The *geometric complexity* (GC) of a function $f : \mathbb{R}^d \to \mathbb{R}^k$ w.r.t. a data distribution $Q = q(x)dx$ on $\mathbb{R}^d$ is defined as the expectation of the gradient Frobenius square-norm w.r.t. to the data distribution [17] given by

$$\mathrm{GC}(f, Q) := \mathbb{E}_{x \sim Q}\left[\|\nabla_x f(x)\|_F^2\right].$$

Intuitively, the GC measures the function complexity or variability and is closely related to the Dirichlet energy in geometric analysis [21]. Provided mild Lipschitz smoothness assumptions [42, Proposition 3.4], the GC can be estimated accurately on batches of training data $D$ by its empirical counterpart

$$\widehat{\mathrm{GC}}(f, D) = \frac{1}{|D|} \sum_{x \in D} \|\nabla_x f(x)\|_F^2, \quad \text{where } D \text{ is an i.i.d. sample drawn from } Q. \tag{4}$$

Previous work has explored the relationship between the GC and model generalization. When measured on the full neural network, lower GC values correlate experimentally with higher test accuracy [45] and it has also been used as a beneficial explicit regularizer [30, 59]. In [16, 17], the authors ignore the softmax activation and study the GC measured with respect to the logit network, exploring its connection with various implicit and explicit regularizers and training heuristics. The logit GC has also be used to prove a margin based multi-class generalization bound [42], assuming that the input distribution $Q$ satisfies a mild assumption known as the Poincaré inequality [1].

In fact, both this work and the proof of the generalization bound rely on the Poincaré inequality, which we argue is indeed a natural and mild assumption for the types of data distributions typically encountered in machine learning; see Appendix A.6 for further discussion. For completeness, we recall it here, cf. [19]. A distribution $Q$ satisfies a Poincaré inequality if, for all differentiable functions $v$ defined on $\mathrm{supp}(Q)$, there exists a constant $c$ such that

$$\mathrm{Var}_v(Q) \leq c\mathbb{E}_{x \sim Q}\left[\|\nabla_x v\|_F^2\right] = c\,\mathrm{GC}(v, Q). \tag{5}$$

Note that the same assumption is used in [7] to prove the law of robustness for overparameterized neural networks via isoperimetric inequalities. In this paper, we peel back yet another layer of the network and consider the embedding geometric complexity measured on a feature map (Section 3).

**Relationship between flatness, neural collapse, and geometric complexity.** The learning-path flatness as measured by its slope influences the geometric complexity of the learned solutions [16, Theorem 5.1]. Namely, for dense layers a regularizing pressure on the slope of the learning path in parameter space transfers to a regularizing pressure on the geometric complexity of the learned solution in function space. A similar result also been shown for special attention layers as well [52].

In this paper, we will see that imposing a regularization pressure on the embedding geometric complexity encourages more neural collapse of the model. This results in the following chain of influences from regularization pressure:

$$\underset{\textit{Regularizing Pressure}}{\text{learning path flatness}} \quad \rightsquigarrow \quad \underset{\textit{Regularizing Pressure}}{\text{function geometric complexity}} \quad \rightsquigarrow \quad \underset{\textit{Lower CDNV}}{\text{embedding neural collapse}}$$

## 3   Problem Formulation

We are interested in understanding the relationship between the geometric complexity (GC) and neural collapse (NC) in the transfer learning setting and explore this relationship in two stages. In the first stage, we examine the general impact of the GC on NC during model training/pre-training. Secondly, we examine how this relationship provides insight into the mechanisms behind transfer learning and the advantageous implicit biases of the pre-training stage. In short, lower GC during pre-training on source classes leads to lower NC for target classes and improved transfer accuracy.

For the first stage of our inquiry (Section 4), we are concerned with a $k$-classification task. Let $D := \{(x_i, y_i)\}_{i=1}^{m}$ be a dataset drawn from a distribution $Q$ with $x_i \in \mathbb{R}^d$ and $y_i \in \{e_j \mid j = 1, \ldots, k\}$ where $e_j \in \mathbb{R}^k$ are the canonical basis vectors representing a one-hot encoding of the labels. We aim to learn this task using a neural network denoted by a function $h_\theta : \mathbb{R}^d \to \mathbb{R}^k$ parameterized by $\theta$ (though to simplify notation we subsequently drop this dependence on $\theta$).

We can write $h_\theta(x) = \text{softmax}(g(f(x))$ where $g : \mathbb{R}^p \to \mathbb{R}^k$ is a classifier head mapping from the feature space $\mathbb{R}^p$ to the prediction layer $\mathbb{R}^k$ and $f : \mathbb{R}^d \to \mathbb{R}^p$ is a feature mapping from the input space to the penultimate embedding layer of the network before the classification head. The standard way of training $h_\theta$ is to find, via some stochastic optimization technique, an optimal parameter configuration $\theta_*$ such that $\theta_* = \text{argmin}_\theta \sum_{i=1}^{m} \ell(h_\theta(x_i), y_i)$ for a given loss function $\ell : \mathbb{R}^k \times \mathbb{R}^k \to [0, \infty)$.

To analyze the role of neural collapse and geometric complexity in this setting, we focus our attention on the feature map $f : \mathbb{R}^d \to \mathbb{R}^p$. The NC as described in Section 2 is defined using this sub-network map $f$ of $h$ and thus we also measure the geometric complexity with respect to this sub-network feature mapping as well. Throughout this paper, we use the term *embedding GC* to refer to the GC of such a feature map $f$ and unless otherwise specified, when we refer to the model GC we mean the embedding GC, not the GC of the logit network as in [16, 17].

Our second stage of inquiry (Section 5) is focused on the role of geometric complexity in transfer learning. For this setting, we assume there is some $l$-class classification target task we would like to solve and corresponding target distribution $\mathcal{T}$. We aim to learn a classifier $h' : \mathbb{R}^d \to \mathbb{R}^l$ given some training data $D' := \{(x_i', y_i')\}_{i=1}^{m'}$ which is potentially very limited in size. In order to find a good solution, we can leverage a pre-trained feature mapping that has been trained on some other source task and source distribution $\mathcal{S}$ where more data is available. For example, by leveraging a feature map $f$ pre-trained as in the above setup, the target task classifier $h'$ can instead be trained on the outputs of our feature map; i.e., $\{f(x_i'), y_i')\}_{i=1}^{m'}$. In this way, we can write $h' = \text{softmax}(g' \circ f)$ where the parameters of $f$ are fixed and we only need to learn the parameters of $g' : \mathbb{R}^p \to \mathbb{R}^l$. Ideally, provided $f$ is a rich enough feature map trained on a general enough source distribution dataset, then the task for learning $g'$ is much simpler.

# 4 Geometric complexity and neural collapse

In this section, we explore the general relationship between the embedding GC and neural collapse showing that the geometric complexity can be used to bound neural collapse. Next, we see how the geometric complexity can be efficiently and accurately estimated, both as an approximation of the theoretical GC as well as through a number of sampling techniques for the empirical GC via batch sampling, feature sampling and label sampling. Lastly, following [22], we derive a new generalization bound for neural networks expressed in terms of the embedding geometric complexity; cf. [42].

## 4.1 Geometric complexity controls neural collapse

Previously, it has been shown [16] that the GC can be controlled implicitly through choice of learning rate and batch size, as well as through standard explicit regularization. Practically, this means that these same beneficial tuning strategies which ensure that models have low GC should also work to keep the NC low as well. The following proposition (which we prove in Appendix A.1) bounds the neural collapse by the geometric complexity provided that the input distribution satisfies a Poincaré inequality also assumed in [42] and [7].

**Proposition 4.1.** *Suppose that we have a balanced multi-class input-distribution $Q$ with $k$ classes satisfying the Poincaré inequality in* (5) *for some constant $c$, then the geometric complexity of a network embedding $f$ bounds its neural collapse as measured by* (3)*; namely, we have the following bound*

$$\mathrm{NC}(f, Q) \leq \frac{c \cdot \mathrm{GC}(f, Q)}{k - 1} \left( \sum_{i \neq j} \frac{1}{d_{ij}^2} \right), \tag{6}$$

*where $k$ is the number of classes, and $d_{ij}$ is the distance between the mean of class $i$ and class $j$.*

We call the RHS of the bound in Eq. (6), excepting the Poincaré constant $c$, the **geometric collapse**:

$$\frac{\mathrm{GC}(f, Q)}{k - 1} \left( \sum_{i \neq j} \frac{1}{d_{ij}^2} \right), \tag{7}$$

where $k$ is the number of classes, and $d_{ij} = \|\mu_f(Q_i) - \mu_f(Q_j)\|$ denotes the Euclidean distance between the mean $\mu_f(Q_i)$ and $\mu_f(Q_j)$ of class $i$ and class $j$ (resp.) in embedding space.

This quantity can be seen as another proxy metric measuring neural collapse. It is made of two main parts decoupling the variances and mean differences present in the neural collapse framework. The variances are consolidated into a single factor through the GC, while the mean differences are averaged across classes in a separate factor. This separation allows the overall intra-class variability to be influenced through the GC term, ensuring sufficient between-class separation through the mechanisms identified in neural collapse.

This bound provides a powerful method to control the overall within-class variability via the L2-norm of the model gradient. As a result, the geometric collapse provides a simplified and more refined approach to managing class separability and variance in deep learning models.

We verify the relationship posed in Eq. (6) through experiments on VGG-13 trained on CIFAR-10 (see Figure 1). Through both implicit and explicit regularizers the pressure on the embedding GC translates to a direct pressure on the neural collapse of the model via the geometric collapse. As already observed in [17], lower levels of GC coincide with higher test accuracy as show in Figure 5 in Appendix A.5 containing the learning curves of that experiment.

In Appendix A.5 we see the same relationship holds across various datasets; e.g., MNIST, Fashion-MNIST, CIFAR-100, and architectures; e.g., ResNet-18, ResNet-50. To avoid possible confounding factors that could arise through batch size and learning rate manipulations, we also directly regularize with the geometric complexity in Appendix A.5.6; the same relations hold in this case too.

## 4.2 The geometric complexity is a reliable and robust measure

One of the key advantages of the GC as a complexity measure is its computational efficiency.

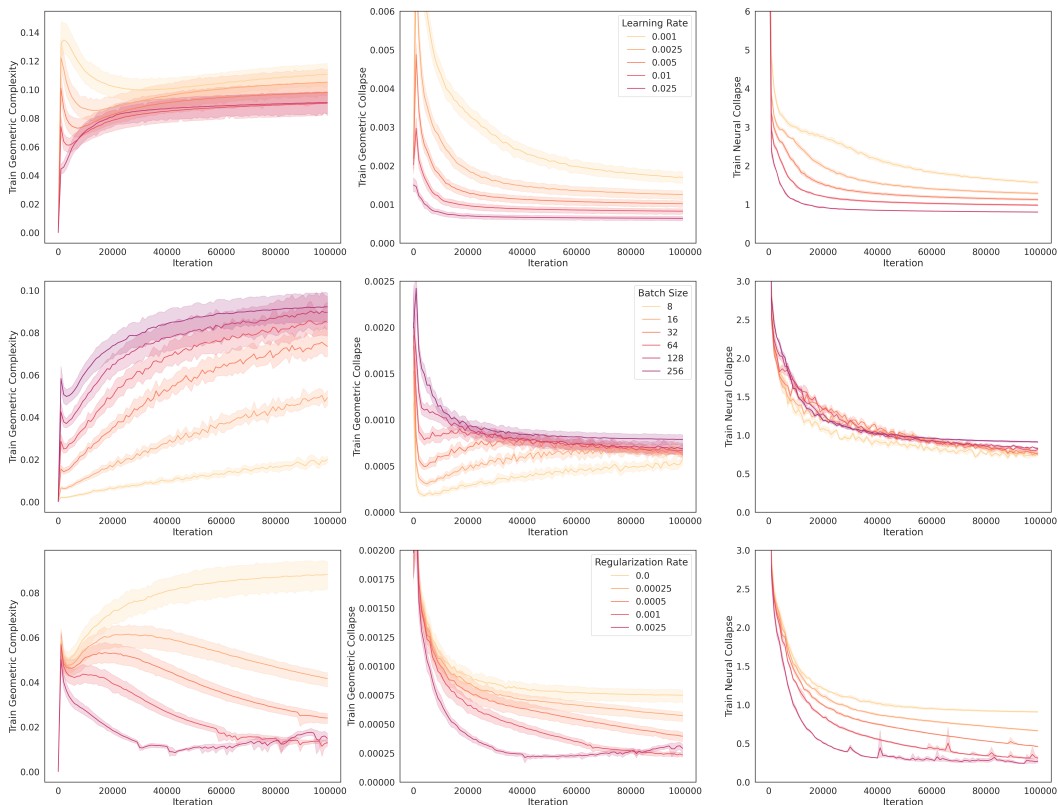

Figure 1: Controlling the neural collapse via the model geometric complexity for VGG-13 trained on CIFAR-10. Lower embedding GC produces lower geometric collapse (Eq. 7) and more neural collapse (i.e., lower NC) for **Top row:** increased learning rates, **Middle row:** decreased batch sizes, and **Bottom row:** increased L2 regularization.

Under mild regularity constraints on the model function the theoretical geometric complexity of a map can be efficiently estimated by its empirical counterpart, as stated in the proposition below, already proven in [42] for logit networks (see Appendix A.2 for a proof). Additionally and crucially, we verify that the empirical GC is a robust and reliable measure with respect to that data sample.

**Proposition 4.2.** *Let $f : \mathbb{R}^d \to \mathbb{R}^p$ be a map with Lipschitz constant $L$ and let $D_X$ be a sample of $m$ elements drawn independently from an input distribution $Q$. Then, for any $\delta > 0$, we have with probability $1 - \delta/2$ the following bound*

$$\mathrm{GC}(f, Q) \leq \widehat{\mathrm{GC}}(f, D_X) + L\sqrt{\frac{\log \frac{2}{\delta}}{2m}}. \tag{8}$$

For a function $f$ and data sample $D_X$ as in Proposition 4.2, to measure $\widehat{\mathrm{GC}}(f, D_X)$ requires computing the Frobenius norm of a potentially very large Jacobian matrix, particularly when $p$ represents the embedding dimension of a feature map. Note that,

$$\widehat{\mathrm{GC}}(f, D_X) = \frac{1}{m} \sum_{i=1}^{m} \sum_{j=1}^{p} \sum_{s=1}^{d} \left( \frac{\partial f^j(x^{(i)})}{\partial x_s} \right)^2. \tag{9}$$

Although the computational complexity of the GC is impervious to increasing complexity of the network architecture, e.g., in terms of parameter count or number of layers, one may run into computation bottlenecks as the sample size $m$ increases or when increasing the dimensionality $d$ (resp. $p$) of the inputs (resp. outputs). In these scenarios, it is necessary to find efficient means to accurately approximate or sample the Jacobian; cf. [61].

We explore the robustness of the GC when measured via samplings along these three axes; i.e., decreasing the number of examples $m$ in the batch, randomly sampling the full Jacobian matrix which has order $d \times p$, and randomly sampling the number of model outputs $p$. As shown Figure 2, we find that the value of the sampled $\widehat{\text{GC}}$ remains stable and consistent to its true value through these fairly simple and naive sampling tricks.

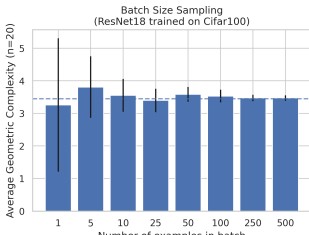 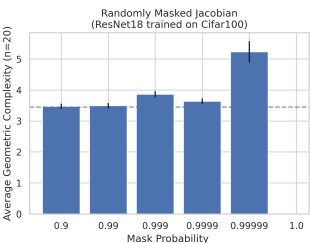 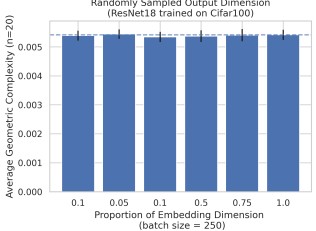

Figure 2: The GC computation is robust and consistent to sampling via **Left:** number of examples in the batch, **Middle:** number of elements in the Jacobian, or **Right:** by sampling the embedding dimension of the model. Here the GC and subnet GC have been computed over 20 trials, plotting the mean and standard deviations for a ResNet-18 model that has been trained to convergence on CIFAR-100. The true value of the GC for each setting is indicated by dotted line.

### 4.3 A new generalization bound with GC through NC

In [42], the authors derive a margin-based multi-class generalization bound for neural networks which depends on the geometric complexity $\text{GC}(h, Q)$ of the full logit network. In this section, we extend this result. With inspiration from [22, Proposition 5], we show that the geometric complexity measured on the sub-network feature map $\text{GC}(f, Q)$ bounds the classification error of the nearest-mean classifier defined by the feature map $f$.

As in [22], given a balanced $k$-class classification problem, let $S = \bigcup_{c \in [k]} S_c$ denote all class samples and define the nearest-mean classifier by $h_{f,S}(x) := \operatorname{argmin}_{c \in [k]} \|f(x) - \mu_f(S_c)\|$ given by the feature map $f$ where $\mu_f(S_c)$ denotes the sample mean of the set $S_c$ under the map $f$. Define the generalization error
$$\text{Error} := \mathbb{E}_{(x,y) \sim P} \left[ \mathbb{I}[h_{f,S}(x) \neq y] \right]$$
where $\mathbb{I}[h_{f,S}(x) \neq y]$ is the indicator function of the error set $[h_{f,S}(x) \neq y]$ and $P$ is the full data distribution from which samples $S$ are drawn.

**Proposition 4.3.** *Suppose that we have a balanced sample $S$ from a $k$-class input-distribution $(x, y) \sim P$ with $m_c$ samples per class. Assume further that the induced input distribution $x \sim Q$ satisfies a Poincaré inequality as in (5) for some constant $c$. Then, for any $\delta > 0$, with probability $1 - \delta/2$ we have the following bound for the generalization error*

$$\mathbb{E}\left[\text{Error}\right] \leq 16c \left( \frac{1}{p} + \frac{1}{m_c} \right) \left( \widehat{\text{GC}}(f, S) + L\sqrt{\frac{\log \frac{2}{\delta}}{2 m_c k}} \right) \left( \sum_{i \neq j} \frac{1}{d_{ij}^2} \right), \tag{10}$$

*where $p$ is the embedding dimension of the feature map $f$ and $d_{ij} = \|\mu_f(S_i) - \mu_f(S_j)\|$. Note, the outer expectation on the left hand side is taken over the samples $S$ used to produce the classifier $h_{f,S}$.*

*Proof.* This follows immediately from [22, Proposition 5] which gives the bound in terms of the neural collapse instead of the geometric complexity. By using Proposition A.1 we can replace the neural collapse with the geometric complexity and using Proposition 4.2 we then replace the theoretical GC with the empirical GC, yielding the additional term with the Poincaré constant. □

In Figure 3, we plot the LHS and RHS of the generalization bound in (10) from Proposition 4.3 for a VGG-13 model trained on CIFAR-10. In this plot, we see that the bound is not vacuous, demonstrating a relatively tight fit. Note that on the RHS, we omitted the term involving the Lipschitz constant $L$, assuming it is small due to its denominator. Additionally, we estimated a lower bound for the Poincaré constant $c$ by comparing the magnitudes of the RHS and LHS in the inequality (6), based on results in Figure 1. This approximation yielded $c \approx 1000$ in our setup.

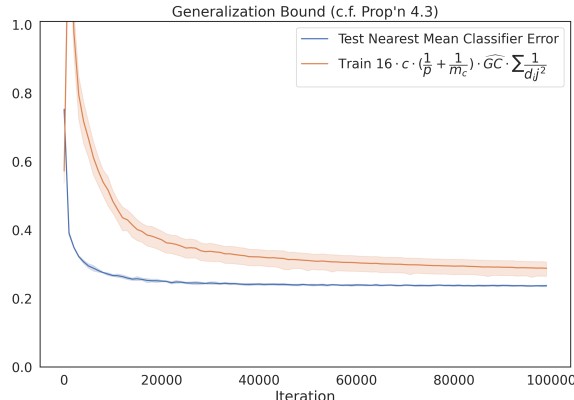

Figure 3: VGG-13 trained on CIFAR-10 with 5 random seeds.

## 5 The impact of geometric complexity on transfer learning

Many implicit regularization mechanisms in gradient descent exert a pressure on geometric complexity which in turn constrain the neural collapse. In this section, we discuss the affect of this implicit bias in transfer learning. Specifically, we show with theory how this bias during pre-training can help explain the mechanisms behind transfer learning. Namely, that lower GC in a pre-trained embedding network promotes neural collapse on new target classes, simplifying the fine-tuning process.

However, our bound makes it clear that certain compatibility conditions between the pre-training (source) distribution and the fine-tuning (target) distribution are needed, which we argue are satisfied in image foundational models and large language models. At last, we verify empirically that known methods to reduce geometric complexity for the pre-trained embedding result in better performance during fine-tuning on unseen tasks, in agreement with our neural collapse bound for transfer learning.

### 5.1 Lower pre-trained GC leads to improved fine-tuning

To analyze the impact of GC in the context of transfer learning, we consider the same formal setup as in [22, Proposition 2]. Assume a class $c$ in a set $\mathcal{C}$ of classes is represented by a class-conditional distribution $Q_c(x) = P(x|y = c)$. Both the pre-training/source classes $\widetilde{Q}_1, \ldots, \widetilde{Q}_k$ and the fine-tuning/target classes $Q_1, \ldots, Q_l$ are assumed to be drawn from a distribution over all classes in $\mathcal{C}$. The induced input distribution on all the classes (i.e., the combined source and target input distribution) is denoted $Q$, while the source-only input distribution is denoted by $\widetilde{Q}$. Let $\mathcal{F}^*$ denote the set of pre-trained feature maps $f : \mathbb{R}^d \to \mathbb{R}^p$ selected by the training procedure (e.g., the set of trained embeddings with different initialization seeds but the same training protocol) and consider

$$\Delta(\mathcal{F}^*) = \inf_{f \in \mathcal{F}^*} \inf_{c \neq c'} \|\mu_f(Q_c) - \mu_f(Q_{c'})\|. \tag{11}$$

With this, we can state the transfer learning bound

**Proposition 5.1.** *Suppose that the source and target input distributions satisfy a Poincaré inequality with constant $c_{\widetilde{Q}}$ and $c_Q$ (resp.). Then, with probability $1 - \delta$, over the selection $Q_c$ and $Q_{c'}$ of target class distributions, we have that the CDNV expectation for two target classes is bounded by the geometric complexity of the embedding network $f$ in the following way*

$$
\begin{aligned}
\mathbb{E}_{Q_c \neq Q_{c'}} \left[ V_f(Q_c, Q_{c'}) \right] \quad \leq \quad & \frac{c_{\widetilde{Q}} \, \mathrm{GC}(f, \widetilde{Q})}{k - 1} \left( \sum_{i \neq j} \frac{1}{d_{ij}^2} \right) \\
& + \left( 8 + \frac{16 c_Q \sup_{c \in \mathcal{C}} \mathrm{GC}(\mathcal{F}^*, Q_c)}{\Delta(\mathcal{F}^*)} \right) \left( \frac{\sqrt{2\pi \log(k)} \mathfrak{H}(\mathcal{F}^*, \widetilde{Q})}{(k - 1) \Delta(\mathcal{F}^*)} \right) \\
& + \left( 1 + \frac{4 \sup_{\substack{x \in \mathrm{supp}(Q) \\ f \in \mathcal{F}^*}} \|f(x)\|}{\Delta(\mathcal{F}^*)^2} \right) \left( \frac{2\sqrt{\log \frac{1}{\delta}} c_Q \sup_{c \in \mathcal{C}} \mathrm{GC}(\mathcal{F}^*, Q_c)}{\sqrt{k} \Delta(\mathcal{F}^*)^2} \right)
\end{aligned}
$$

*where $k$ is the number of source classes,* $\mathrm{GC}(\mathcal{F}^*, Q_c) = \sup_{f \in \mathcal{F}^*} \mathrm{GC}(f, Q_c)$, $d_{ij} = \|\mu_f(Q_{c_i}) - \mu_f(Q_{c_j})\|$, *and* $\mathfrak{H}(\mathcal{F}^*, \widetilde{Q})$ *is a complexity measure for* $\mathcal{F}^*$ *over* $\widetilde{Q}$ *(see Appendix A.3).*

*Proof.* This follows immediately from [22, Proposition 2] which proves this bound for the NC and distribution variances. Using our Proposition 4.1, we can replace the NC by the GC and swap the variances by the corresponding geometric complexities via the respective Poincaré inequality. $\square$

**Interpretation.** In the bound above, the first term depends on the geometric complexity measured over the source distribution $\tilde{Q}$ and decreases as the number of source classes $k$ increases. Thus, we can predict that lower $\mathrm{GC}(f, \tilde{Q})$ encourages lower NC values (i.e. more neural collapse) on the new target classes provided that the second and third term of the bound are also small.

However these other two terms involve both the source and target classes, making the full bound no longer dependent only on the geometric complexity over $\tilde{Q}$.

Moreover these two terms can apriori explode since the infimum over the class-mean distances $\Delta(\mathcal{F}^*)$ no longer only involves source classes (where this distance is bounded away from zero due to neural collapse into an equidistant simplex) but also target classes (for which we no longer have theoretical guarantees).

However, if the source labels are granular enough in the sense the target labels can be represented as combinations of the source labels (as for instance a target label of "dog" can be subsumed as all the breeds of dogs in source classes on ImageNet), then the issues above are mitigated. Formally, we can summarize this granularity compatibility condition between the source and target classes as follows: For each label $c$ of the target class there exists labels $c_1, \ldots, c_k$ in the source classes such that $Q_c = 1/k(Q_{c_1} + \cdots + Q_{c_k})$. Because $\mathrm{GC}(f, \cdot)$ is linear, we have

$$\sup_{c \in \mathcal{C}} \mathrm{GC}(\mathcal{F}^*, Q_c) \leq \sup_{c \in [k]} \mathrm{GC}(\mathcal{F}^*, \tilde{Q}_c)$$

and thus recover dependence only on source classes.

To address the $\Delta(\mathcal{F}^*)$ term, observe that the compatibility condition above implies in terms of class-means that $\mu_f(Q_c) = \frac{1}{k}(\mu_f(Q_{c_1}) + \cdots + \mu_f(Q_{c_k}))$. Because of neural collapse during pre-training, these source class-means form the vertices of a face in the neural-collapse class-mean simplex making the target-class mean $\mu_f(Q_c)$ the barycenter of this face. Since the distance between any two points taken in the vertices of a simplex plus its barycenters is bounded away from zero, so is $\Delta(\mathcal{F}^*)$. Note, albeit intuitive and natural, this granularity condition is hard to verify in practice.

## 5.2 Improve fine-tuning by controlling pre-trained GC

When the compatibility conditions above are satisfied, Proposition 5.1 indicates that increasing the amount of neural collapse for the target classes can be achieved by lowering the embedding GC during pre-training. We verify this indeed occurs experimentally using the same regularization techniques exploited in Section 4. Furthermore, we verify that these implicit methods of controlling pre-training GC produce feature maps that perform better on fine-tuning tasks on CIFAR-FS with a ResNet-18 in Figure 4 and on mini-ImageNet with VGG-16 in Appendix A.5.5.

## 6 Limitations and Conclusion

There is a notable limitation when extending our findings to language modeling and, for example, large language models (LLMs). However, this limitation is not specific to our work per-se, but instead it is a limitation of the application of neural collapse to language models in general, as described in the recent work [65]. Namely, language modeling, as conducted via training by token prediction, creates a classification task where the conditions for neural collapse are implausible. The main problem, in addition to an imbalanced vocabulary, is that the embedding dimension for language models is typically far less than the number of classes (i.e., the total vocabulary size), making the neural collapse simplex impossible to exist. Extending the notion of neural collapse to the language models is an open question and an active area of research and beyond the scope of this work.

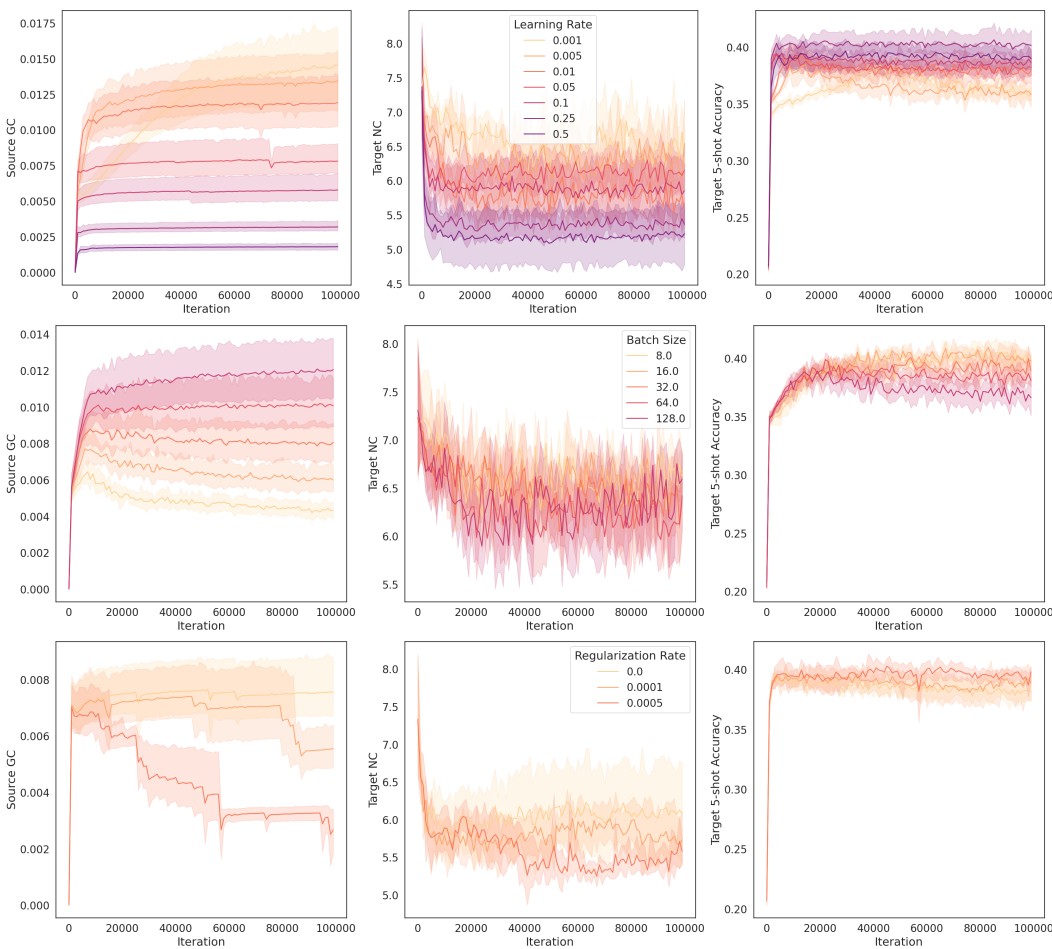

Figure 4: Controlling target neural collapse through source GC on CIFAR-FS with ResNet-18. Lower Source GC produces more neural collapse on target classes (i.e. lower target NC), and higher 5-shot transfer accuracy for **Top row:** increased learning rates, **Middle row:** decreased batch sizes, and **Bottom row:** increased L2 regularization.

Uncovering and understanding the implicit biases that enable successful transfer learning is a critical area of research in modern machine learning. Here we provide a framework connecting three different themes behind implicit regularization: flatness of the loss surface, geometric complexity, and neural collapse. We show that the embedding geometric complexity directly controls the neural collapse during training and, moreover, plays a role in the success of transfer learning.

Extensive experiments on different image classification and fine tuning tasks across different hyperparameters verify our hypothesis. We believe this opens up an intriguing direction of research further exploring the role of geometric collapse and geometric complexity in deep learning and provides valuable insight for designing more efficient and effective techniques for model training and fine-tuning.

## Acknowledgments and Disclosure of Funding

We would like to thank Patrick Cole for their support. We would also like to thank Hanna Mazzawi and Peter Bartlett for helpful conversations.

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

# A  Appendix

## A.1  Proof of Proposition 3.2 (Neural collapse bound)

*Proof.* Suppose that we have an input probability distribution $Q(x) = q(x)dx$ coming from a data distribution $P(X, Y)$ (i.e. $q(x)dx$ is the marginal probability distribution of $p(x, y)dxdy$) where $Y$ represents the $k$ possible classes. Suppose further that $P(X, Y)$ is a balanced multi-class distribution. In terms of the input distribution $Q(x)$ this means that

$$Q = \frac{1}{k}(Q_1 + \cdots + Q_k)$$

where $Q_i(x) = q_i(x)dx$ is the input distribution of class $i$. Suppose moreover that $Q(x)$ satisfies the Poincaré inequality in (5), that is,

$$\mathrm{Var}_f(Q) \leq c\mathbb{E}_Q(\|\nabla_x f\|_F^2) = c\,\mathrm{GC}(f, Q)$$

for some constant $c$. Then the statement we want to prove is that the geometric complexity of $f$, that is,

$$\mathrm{GC}(f, Q) := \int \|\nabla_x f(x)\|^2 q(x)dx$$

bounds its neural collapse as measured by

$$\mathrm{NC}(f, Q) := \frac{1}{\#\{i \neq j\}} \sum_{i \neq j} \left( \frac{\mathrm{Var}_f(Q_i) + \mathrm{Var}_f(Q_j)}{2\|\mu_f(Q_i) - \mu_f(Q_j)\|^2} \right) \tag{12}$$

More precisely, we want to prove that we have the following inequality:

$$\mathrm{NC}(f, Q) \leq \frac{c \cdot \mathrm{GC}(f, Q)}{k - 1} \left( \sum_{i \neq j} \frac{1}{d_{ij}^2} \right), \tag{13}$$

where $d_{ij} = \|\mu_f(Q_i) - \mu_f(Q_j)\|$ is the distance between the mean of class $i$ and class $j$. Let us now prove this statement.

First of all, since the geometric complexity respects convex sums of data densities, we have that for a distribution $Q = \frac{1}{k}(Q_1 + \cdots Q_k)$ we can write

$$\mathrm{GC}(f, Q) = \frac{1}{k}(\mathrm{GC}(f, Q_1) + \cdots + \mathrm{GC}(f, Q_k)). \tag{14}$$

In particular, this means that $\mathrm{GC}(f, Q_i) \leq k\,\mathrm{GC}(f, Q)$. Furthermore, the Poincare inequality ensures that $\mathrm{Var}_{Q_i}(f) \leq c\,\mathrm{GC}(f, Q_i)$. Using these two properties and the definition of the neural collapse, we obtain that:

$$\begin{aligned}
\mathrm{NC}(f, Q) &= \frac{1}{\#\{i \neq j\}} \sum_{i \neq j} \left( \frac{\mathrm{Var}_f(Q_i) + \mathrm{Var}_f(Q_j)}{2\|\mu_f(Q_i) - \mu_f(Q_j)\|^2} \right) \\
&\leq \frac{c}{\#\{i \neq j\}} \sum_{i \neq j} \frac{\mathrm{GC}(f, Q_i) + \mathrm{GC}(f, Q_j)}{2d_{ij}^2} \\
&\leq \frac{ck\,\mathrm{GC}(f, Q)}{k(k - 1)} \left( \sum_{i \neq j} \frac{1}{d_{ij}^2} \right),
\end{aligned}$$

which simplifies to the desired result. $\qquad\square$

**Remark A.1.** *Excepting the Poincaré constant, the quantity on the RHS above given by*

$$\frac{\mathrm{GC}(f, Q)}{k - 1} \left( \sum_{i \neq j} \frac{1}{d_{ij}^2} \right) \tag{15}$$

*could be taken as an alternative measure of neural collapse. We call this the **geometric collapse**.*

## A.2 Proof of Proposition 4.2 (Estimating the Theoretical GC by $\widehat{\text{GC}}$)

*Proof.* Let $Q$ be an (input) probability distribution defined over $\mathbb{R}^d$ and let $D = \{x_i\}_{i=1}^m$ of $m \geq 1$ points drawn as i.i.d. samples from $Q$. Given the map $f : \mathbb{R}^d \to \mathbb{R}^p$, we denote the empirical geometric complexity of $f$ over $D$ by $\widehat{\text{GC}}(f, D)$ which is defined as

$$\widehat{\text{GC}}(f, D) = \frac{1}{m} \sum_{i=1}^m \|\nabla_x f(x_i)\|_F^2.$$

We begin by showing that

$$\mathbb{E}_{D \sim Q^m}\left[\widehat{\text{GC}}(f, D)\right] = \text{GC}(f, Q).$$

This follows by direct computation, keeping in mind that $Q$ is a probability distribution and that the points are independently sampled. We can write $Q$ as $Q(x) = q(x)dx$ and note that

$$
\begin{aligned}
\mathbb{E}_{D \sim Q^m}\left[\widehat{\text{GC}}(f, D)\right] &= \mathbb{E}_{x_1, \ldots, x_m \sim Q^m}\left[\frac{1}{m} \sum_{i=1}^m \|\nabla_x f(x_i)\|_F^2\right] \\
&= \frac{1}{m} \int_{\mathbb{R}^{m \times d}} \sum_{i=1}^m \|\nabla_x f(x_i)\|_F^2 dQ^m(x_1, \ldots, x_m) \\
&= \frac{1}{m} \int_{\mathbb{R}^{m \times d}} \sum_{i=1}^m \|\nabla_x f(x_i)\|_F^2 q(x_1) \cdots q(x_m) dx_1 \cdots dx_m \\
&= \frac{1}{m} \sum_{i=1}^m \int_{\mathbb{R}^{(m-1) \times d}} \left[\int_{\mathbb{R}^d} \|\nabla_x f(x_i)\|_F^2 q(x_i) dx_i\right] q(x_1) \cdots \widehat{q(x_i)} \cdots q(x_m) dx_1 \cdots \widehat{dx_i} \cdots dx_m \\
&= \frac{1}{m} \sum_{i=1}^m \left[\int_{\mathbb{R}^d} \|\nabla_x f(x_i)\|_F^2 q(x_i) dx_i\right] \\
&= \frac{1}{m} \sum_{i=1}^m \left[\int_{\mathbb{R}^d} \|\nabla_x f(x_i)\|_F^2 dQ(x_i)\right] \\
&= \frac{1}{m} \sum_{i=1}^m \text{GC}(f, Q) \\
&= \text{GC}(f, Q).
\end{aligned}
$$

Now suppose we have two separate independent samples $D$ and $D'$ each of of size $m \geq 1$ which differ by exactly one point, say $x_i$ in $D$ and $x_i'$ in $D'$. By assumption, the map $f$ is $L$-Lipschitz for some Lipschitz constnt $L > 0$. Therefore, we have

$$\widehat{\text{GC}}(f, D) - \widehat{\text{GC}}(f, D') = \frac{1}{m}\left(\|\nabla_x f(x_i)\|_F^2 - \|\nabla_x f(x_i')\|_F^2\right) \leq L^2/m,$$

and similarly, $\widehat{\text{GC}}(f, D') - \widehat{\text{GC}}(f, D) \leq L^2/m$. It follows that $|\widehat{\text{GC}}(f, D) - \widehat{\text{GC}}(f, D')| \leq L^2/m$ and by applying McDiarmind's inequality (e.g., [41]), we have that for any $\epsilon > 0$,

$$\mathbb{P}\left[\widehat{\text{GC}}(f, D) - \mathbb{E}_{D \sim \mu^m}[\widehat{\text{GC}}(f, D)] \leq \epsilon\right] \geq 1 - \exp(-2m\epsilon^2/L^2). \tag{16}$$

Since, from the computation above, $\mathbb{E}_{D \sim \mu^m}[\widehat{\text{GC}}(f, D)] = \text{GC}(f, Q)$ and setting $\delta/2 = \exp(-2m\epsilon^2/L^2)$ and substituting for $\epsilon$ in (16), we get that for any $\delta > 0$ with probability as least $1 - \delta/2$ the following holds:

$$\text{GC}(f, Q) \leq \widehat{\text{GC}}(f, D) + L\sqrt{\frac{\log \frac{2}{\delta}}{2m}}.$$

This completes the proof. $\qquad\square$

## A.3 Definition of the complexity measure $\mathfrak{H}(\mathcal{F}, Q)$ in Proposition 4.1

Let us restate here for the sake of completeness the definition given in Galanti et al. [22]. First consider the Rademacher complexity $R(A)$ of a set $A \subset \mathbb{R}^n$ which is given as the expectation $\mathbb{E}_\epsilon(X)$ of the random variable $X = \sup_{a \in A} \langle \epsilon, a \rangle$ where $\epsilon = (\epsilon_1, \ldots, \epsilon_n)$ is a random vector with components uniformly distributed among the two values -1 and 1. Now consider the input distribution $Q$ coming from a multiclass distribution with labels in $c \in \mathcal{C}$ with label input distribution $Q_c$. We define the complexity measure $\mathfrak{H}(\mathcal{F}, Q)$ for the candidate functions in $\mathcal{F}$ over the source distribution $Q$ as the Rademacher complexity of the following set:

$$\{(\mu_f(Q_c), \text{Var}_f(Q_c)) : c \in \mathcal{C}, f \in \mathcal{F}\}.$$

## A.4 Experiment details

### A.4.1 Figure 1: Geometric Complexity Controls Neural Collapse on CIFAR-10.

We trained a VGG-13 neural network on the full CIFAR-10 dataset with the provided architecture [56] and using the standard train/test split. Throughout training, we reported the following metrics measured and averaged over multiple batches of the training dataset: 1) the geometric complexity of the model embedding layer; i.e., the layer on which the neural collapse is measured, 2) the neural collapse as measured by (3) measured on the model embedding layer, and 3) the geometric collapse of the model also measured on the penultimate embedding layer of the model and given by

$$\frac{\text{GC}(f, Q)}{k - 1} \left( \sum_{i \neq j} \frac{1}{d_{ij}^2} \right) \tag{17}$$

where $Q$ is the training distribution, $k = 10$ as this was CIFAR-10, and (as in the paper) where $d_{ij} = \|\mu_f(Q_i) - \mu_f(Q_j)\|$ is the distance between the sample means of class $i$ and class $j$.

These quantities were measured every 1000 steps of a very long pre-training of $100,000$ steps. The optimizer was plain SGD (note we obtained similar results with momentum) without any regularization nor schedule to avoid masking effects. We used random crop and random flip for data augmentation. The results were all averaged over 5 random seeds for the neural network parameter default initialization. The plots have no smoothing applied to the learning curves. **Top row:** We swept over a learning rate range of $\{0.001, 0.0025, 0.005, 0.01, 0.025, 0.1\}$ with a constant batch size of 512. **Middle row:** We swept over a batch size range of $\{8, 16, 32, 64, 128, 256\}$ with a constant learning rate of 0.01. **Bottom row:** We swept over a L2 regularization rate range of $\{0.0, 0.00025, 0.0005, 0.001, 0.0025\}$ with learning rate 0.01 and batch size 256. Each sweep took roughly 10h of training on a single Google Cloud TPU V3 accessed via a Google colab.

In Figure 5, we show the complete learning curves of that experiment.

### A.4.2 Figure 3: Tightness of the generalization bound on CIFAR-10

We trained a VGG-13 neural network [56] on the full CIFAR-10 dataset using the provided train/test split. For this model architecture we use an embedding layer dimension $p = 1024$. Note also that the number of examples per class is $m_c = 5000$. We used a constant learning rate of $0.005$, a batch size of 512 and trained for 100000 steps reporting metrics every 1000 steps.

Throughout training, we reported the following metrics measured and averaged over multiple batches of the training dataset: 1) the empirical geometric complexity of the model embedding layer; i.e., the layer on which the neural collapse is measured, denoted $\widehat{\text{GC}}$, and 2) the sum of the inverse squares of $d_{ij}$ which denotes the distance between the mean of class $i$ and class $j$ for $i, j \in [k]$ for $i \neq j$ and, here, $k = 10$. These quantities were used to create the blue curve in Figure 3. We also computed the nearest mean classifier error on the test set. Recall, for the feature map $f$ and a sample $S$ the nearest mean classifier is defined as $h_{f,S} := \text{argmin}_{c \in [k]} \|f(x) - \mu_f(S_c)\|$ where, here $k = 10$, and $\mu_f(S_c)$ denotes the sample mean of the set $S_c$ for class $c$ under the feature map $f$. To create the orange curve in Figure 3 we plot the average error $h_{f,S}(x) \neq y$ over the test set.

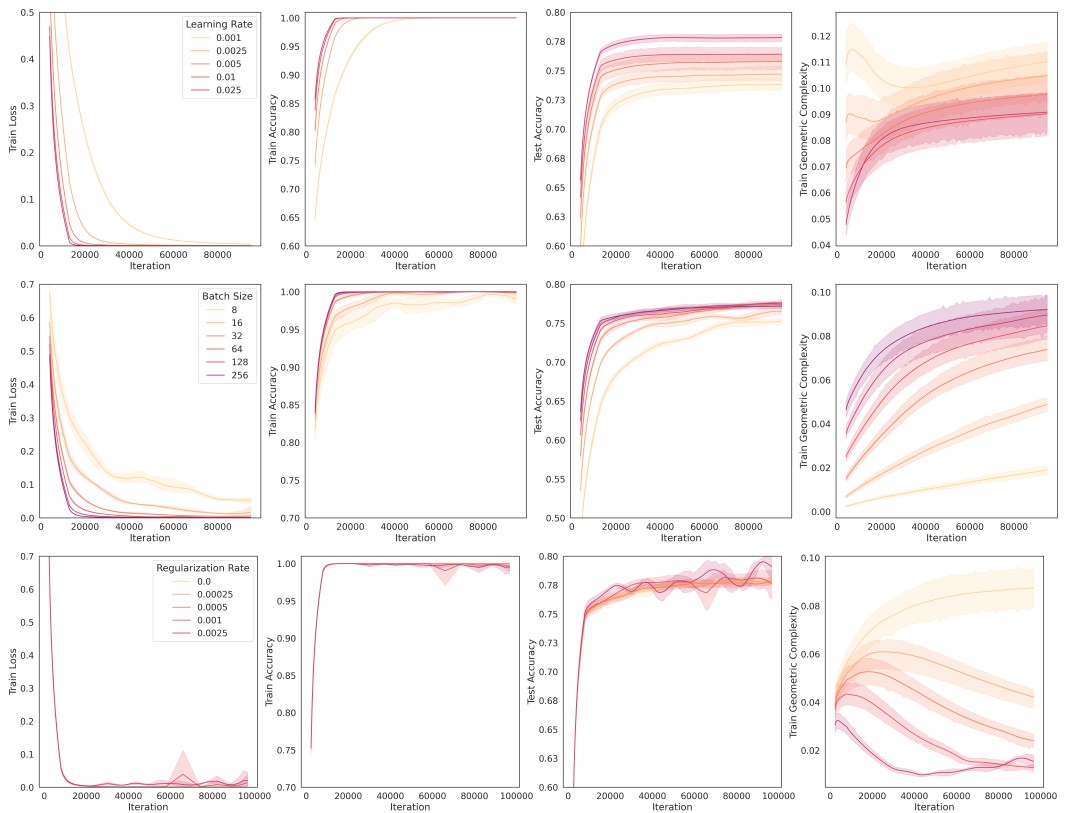

Figure 5: VGG-13 on CIFAR-10. Training has reached terminal phase of training (TPT) with training accuracy equal to 1 (see **first** and **second** column). Lower GC in training correlates with higher test accuracy in all settings (see **third** and **fourth** column).

### A.4.3 Figure 4: Controlling target performance through source GC on CIFAR-FS.

We trained a RestNet-18 neural network with width 1 implemented in Flax `https://github.com/google/flax/blob/main/examples/imagenet/models.py` on CIFAR-FS with its initial convolution modified to stride 1 and kernel size of 3 to adapt to CIFAR-FS instead of ImageNet. We used only 10 classes for the source datasets with 600 images per class, which further we split in a 10/90 split for the test and train splits. The metrics were computed on an average of 100 random samples of 5 classes in the remaining classes of CIFAR-FS. We reported 1) the geometric complexity of the embedding layer measured on the source dataset (Source GC), 2) the neural collapse as measured by (3) measured and averaged on the 100 random samples of 5-label samples from the target dataset, 3) the test accuracy obtained by solving the normal equation directly for a ridge regression with only 5 examples per class obtained from the target set. These three quantities were measured every 1000 steps of a very long pre-training of $100_000$ steps. The optimizer was plain SGD (note we obtained similar results with momentum) without any regularization nor schedule to avoid masking effects. We used random crop and random flip for data augmentation. The results were all averaged over 5 random seeds for the neural network parameter default initialization. The plots have no smoothing applied to the learning curves. **Top row:** We swept over a learning rate range of $\{0.001, 0.005, 0.01, 0.05, 0.1, 0.25, 0.5\}$ with a constant batch size of 256. **Middle row:** We swept over a batch size range of $\{8, 16, 32, , 64, 128\}$ with a constant learning rate of 0.005. **Bottom row:** We swept over a L2 regularization rate range of $\{0, 0.0001, 0.0005\}$ Each sweep took roughly 10h of training on a single Google Cloud TPU V3 accessed via a Google colab.

## A.5 Additional Experiments

### A.5.1 Cifar-10 on ResNet-18

In Figure 6, we trained a ResNet-18 neural network on CIFAR-10. The experimental setup is the same as described in Appendix A.4.1. The results were all averaged over 5 random seeds for the neural network parameter default initialization. The plots have no smoothing applied to the learning curves.

**Top row:** We swept over a learning rate range of $\{0.005, 0.01, 0.025, 0.05, 0.1, 0.2\}$ with a constant batch size of 256. **Middle row:** We swept over a batch size range of $\{8, 16, 32, 64, 128, 256\}$ with a constant learning rate of 0.2. **Bottom row:** We swept over a L2 regularization rate range of $\{0.0, 0.0001, 0.00025, 0.0005, 0.00075\}$ with learning rate 0.02 and batch size 512.

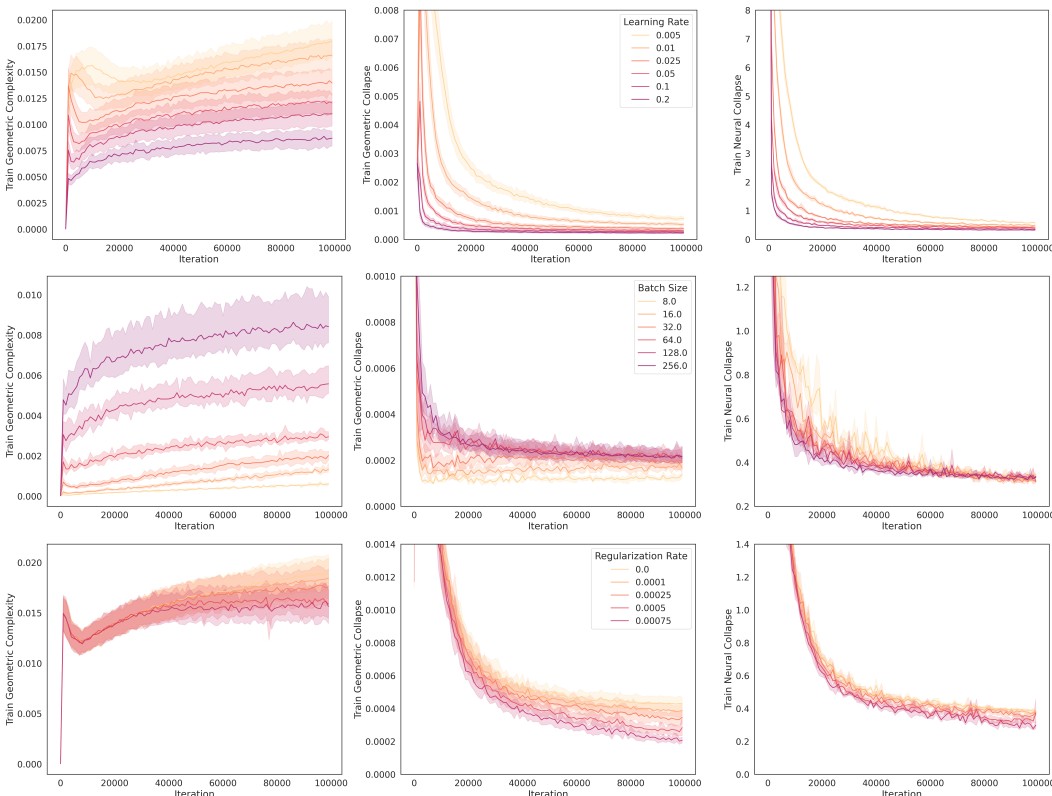

Figure 6: Controlling the neural collapse via the model geometric complexity for ResNet-18 trained on CIFAR-10. Lower GC produces lower geometric collapse and more neural collapse (i.e., lower NC) for **Top row:** increased learning rates, **Middle row:** decreased batch sizes, and **Bottom row:** increased L2 regularization.

### A.5.2 MNIST on VGG-11

In Figure 7, we trained a VGG-11 neural network [56] on MNIST. The experimental setup is the same as described in Appendix A.4.1. The results were all averaged over 5 random seeds for the neural network parameter default initialization. The plots have no smoothing applied to the learning curves.

**Top row:** We swept over a learning rate range of $\{0.0025, 0.005, 0.01, 0.025, 0.05, 0.1\}$ with a constant batch size of 512. **Middle row:** We swept over a batch size range of $\{8, 16, 32, 64, 128, 256, 512, 1024\}$ with a constant learning rate of 0.005. **Bottom row:** We swept over a L2 regularization rate range of $\{0, 0.00001, 0.0001, 0.00025, 0.0005\}$ with learning rate 0.02 and batch size 512.

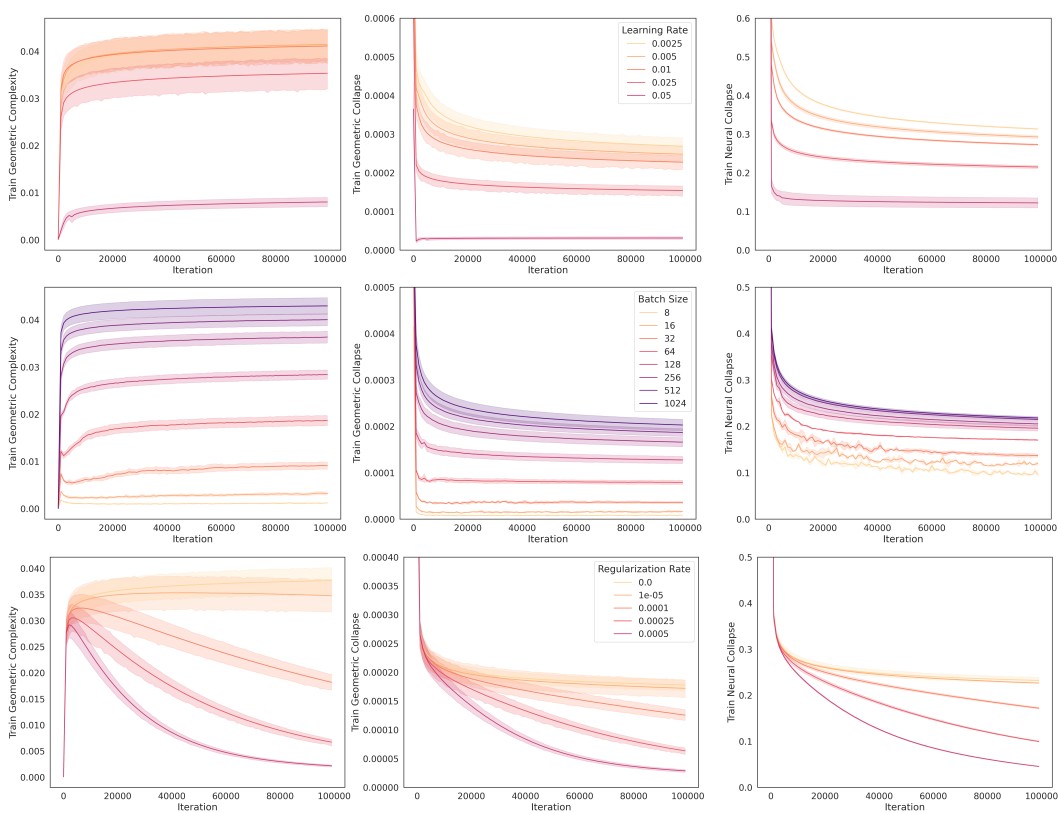

Figure 7: Controlling the neural collapse via the model geometric complexity for VGG-11 trained on MNIST. Lower GC produces lower geometric collapse and more neural collapse (i.e., lower NC) for **Top row:** increased learning rates, **Middle row:** decreased batch sizes, and **Bottom row:** increased L2 regularization.

### A.5.3  Fashion-MNIST on VGG-11

In Figure 8, we trained a VGG-11 neural network [56] on Fashion-MNIST. The experimental setup is the same as described in Appendix A.4.1. The results were all averaged over 5 random seeds for the neural network parameter default initialization. The plots have no smoothing applied to the learning curves.

**Top row:**  We swept over a learning rate range of $\{0.005, 0.01, 0.025, 0.05, 0.1, 0.2\}$ with a constant batch size of 256.  **Bottom row:**  We swept over a L2 regularization rate range of $\{0, 0.00001, 0.0001, 0.00025, 0.0005\}$ with learning rate 0.02 and batch size 512.

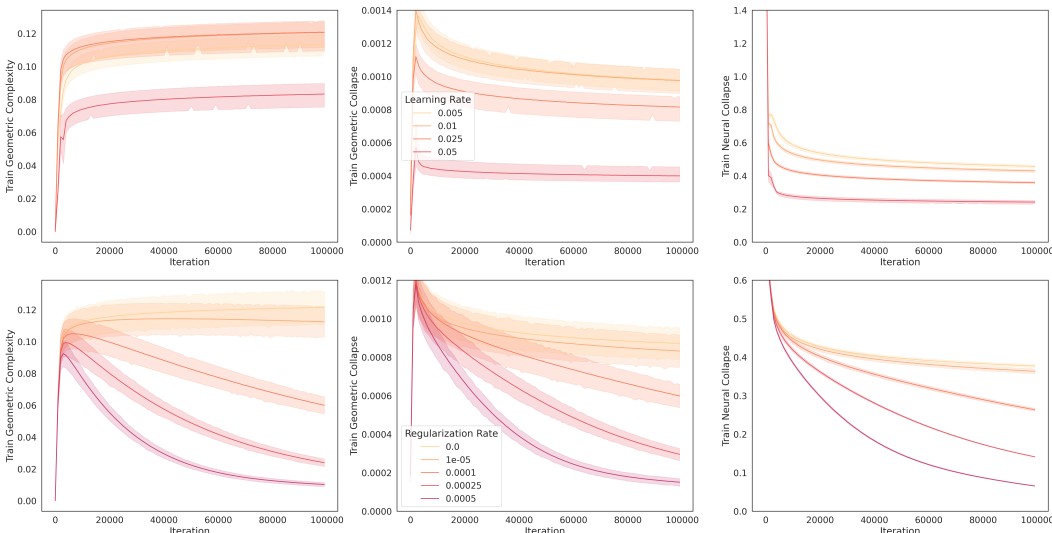

Figure 8: Controlling the neural collapse via the model geometric complexity for VGG-11 trained on Fashion-MNIST. Lower GC produces lower geometric collapse and more neural collapse (i.e., lower NC) for **Top row:** increased learning rates, **Bottom row:** increased L2 regularization.

### A.5.4  Cifar-100 on ResNet50

In Figure 9, we trained a ResNet-50 neural network on CIFAR-100. The experimental setup is the same as described in Appendix A.4.1. The results were all averaged over 5 random seeds for the neural network parameter default initialization. The plots have no smoothing applied to the learning curves.

**Top row:**  We swept over a learning rate range of $\{0.005, 0.01, 0.05, 0.1, 0.2\}$ with a constant batch size of 256.  **Bottom row:**  We swept over a L2 regularization rate range of $\{0.0, 0.0001, 0.00025, 0.0005, 0.00075\}$ with learning rate 0.02 and batch size 256.

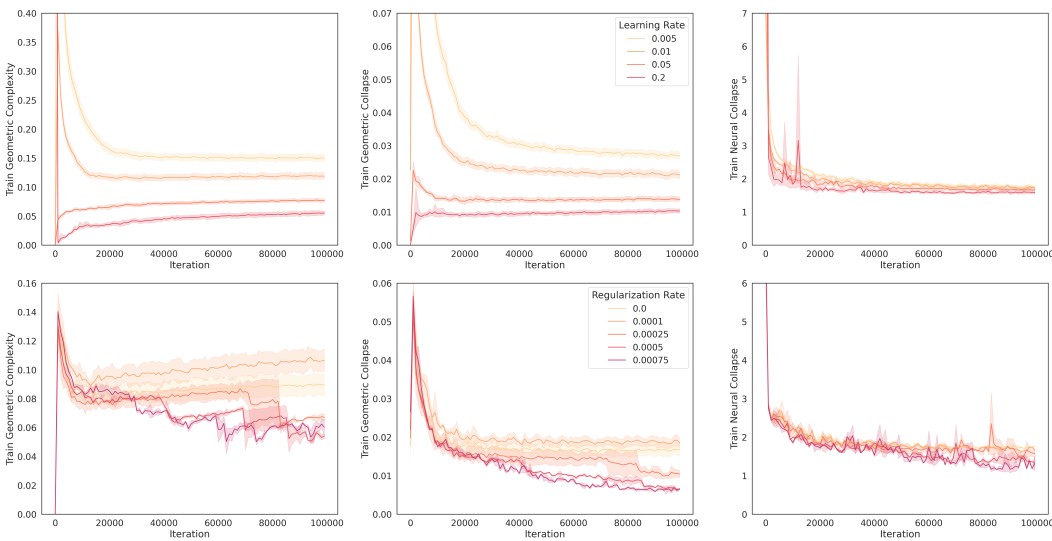

Figure 9: Controlling the neural collapse via the model geometric complexity for ResNet-50 trained on CIFAR-100. Lower GC produces lower geometric collapse and more neural collapse (i.e., lower NC) for **Top row:** increased learning rates, **Bottom row:** increased L2 regularization.

### A.5.5 Lower Pre-trained GC Leads to Improved Fine-tuning: mini-ImageNet dataset on VGG architecture

In Figure 10, we trained a VGG16 neural network (as described in [57]) implemented in Flax on Mini-ImageNet (as described in [60]). We used 10 classes with 600 examples per class for training, with a 10/90 train/test split. To evaluate the downstream performance we used 100 downstream tasks consisting of 5 labels for few shot learning randomly chosen from a pool of 20 separate from the training labels. The experimental setup was the same as described in section A.4.3. Namely, we reported 1) the geometric complexity of the embedding layer measured on the source dataset (Source GC), 2) the neural collapse as measured by (3) measured and averaged on the 100 random samples of 5-label samples from the target dataset, 3) the test accuracy obtained by solving the normal equation directly for a ridge regression with only 5 examples per class obtained from the target set. These three quantities were measured every 1000 steps of a very long pre-training of 100,000 steps. The optimizer was plain SGD (note we obtained similar results with momentum) without any regularization nor schedule to avoid masking effects. We used random crop and random flip for data augmentation. The results were all averaged over 5 random seeds for the neural network parameter default initialization. We swept over a learning rate range of $\{0.001, 0.005, 0.01\}$ with a constant batch size of 256. Each sweep took roughly 10h of training on a single Google Cloud TPU V3 accessed via a Google colab.

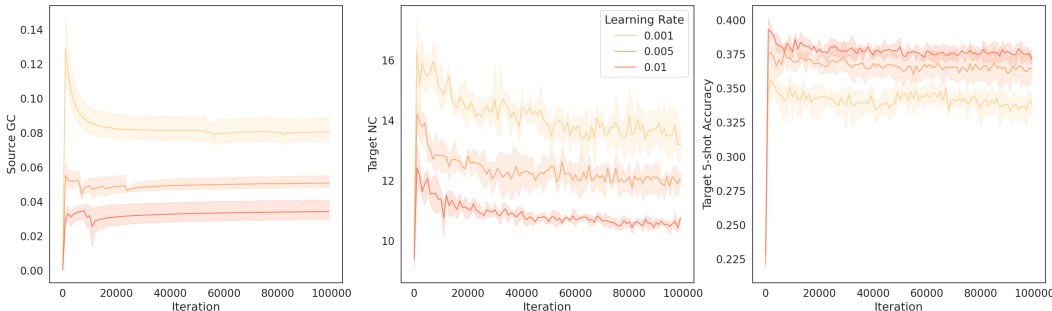

Figure 10: Controlling target NC through source GC on mini-imagenet with VGG-16: Increased learning rates produces lower GC and more neural collapse (i.e. lower NC) resulting in higher 5-shot transfer accuracy.

### A.5.6 Direct GC regularization increases neural collapse

To mitigate possible confounding factors due to batch size and learning rate manipulations, we train a VGG-13 model [57] on CIFAR-10 with explicit GC regularization using a fixed learning rate of 0.01 and a batch size of 256. We explore increasing levels of explicit GC regularization with rates $10^{-6}, 10^{-5}, 10^{-4}$ and plot the results in Figure 11. We observe the same predictions as in our experiments which indirectly lower the GC through learning rates, batch sizes and L2 regularization. Namely, lower GC produces lower NC values (i.e., increased neural collapse). Note that in this experiment we regularized w.r.t. the GC computed at the logit layer rather than the embedding GC. This is because the high dimension of the embedding layer makes taking the gradient of the embedding GC prohibitively expensive. Already, regularization with GC taken at the logit layer was possible only for one seed. That being said, explicit regularization by GC computed at the logit layer provides a direct way to control GC computed at the embedding layer.

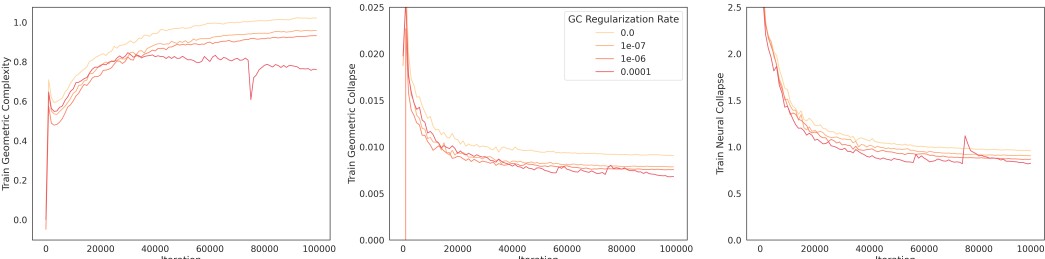

Figure 11: VGG-13 on CIFAR-10 with explicit GC regularization (one seed).

### A.6 Discussion on the Poincaré inequality

Recall that a distribution $Q$ satisfies a Poincaré inequality if, for all differentiable functions $v$ defined on the support of $Q$, there exists a constant $c > 0$ such that

$$\mathrm{Var}_v(Q) \le c\mathbb{E}_{x \sim Q}\left[\|\nabla_x v\|_F^2\right]. \tag{18}$$

In this section, we argue that the Poincare Inequality (PI) is a mild and natural assumption to make. For instance, the Gaussian distribution, mixtures of Gaussians with equal (or different) variances, mixtures of Gaussians and sub-Gaussians, mixtures of uniform and Gaussian measures; and any log-concave probability measure all satisfy a PI; see, for example, [1, 54]. The same is true for most distributions of importance in probability and statistics; see [47].

The Poincaré constant $c$ itself can take very high values depending on the spread of the distribution. For example, for a standard normal distribution the Poincare constant is 1 while for a multivariate normal distribution the Poincare constant equals the largest eigenvalue of its covariance matrix which can be any positive real number. Intuitively speaking, the Poincaré constant will increase if the space is stretched in any direction and decrease if it is compressed in any direction.

The PI has also been assumed to hold for real life image datasets, where it has been used to help improve GAN convergence, as in [31]. It is also a key assumption to understand the role of over-parameterization in generalization as happens for large neural networks (cf. [7] which frames this as an equivalent isoperimetry condition).

On the contrary, non-PI distributions are considered pathological; they can be constructed for instance by artificially concatenating distributions with fully disjoint support. See [42] for a construction of pathological examples of distributions not satisfying a Poincaré inequality.

