# OpenReview forum: "The Impact of Geometric Complexity on Neural Collapse in Transfer Learning"
_NeurIPS.cc/2024/Conference — NeurIPS 2024 poster_

### Official Review · Reviewer_pQCH · 2024-07-06

**Soundness:** 3
**Presentation:** 3
**Contribution:** 2
**Rating:** 5
**Confidence:** 4

**Summary:**

The paper studies the relationship between geometric complexity (based on penultimate layer features) and the neural collapse phenomenon (especially, the variability collapse property). The main theoretical result is a bound on the CDNV-based NC metric using geometric complexity (Proposition 4.1). This is then employed to extend generalization and transfer learning bounds (proposed by previous efforts) to rely on geometric complexity. Experimental results on classification tasks (along with transfer learning settings) are presented to validate the proposed relationship.

**Strengths:**

The unified analysis of neural collapse and geometric complexity presents an interesting line of research for the community. Especially, in terms of studying seemingly disjoint phenomena under a common lens.

**Weaknesses:**

1. There seems to be a mismatch in the $\log(2/\delta)$ term in Proposition 4.2 vs $\log(1/\delta)$ term in Proposition 4.3, which makes the main results inconsistent (yet fixable). Additionally, why do we have an "expectation" over the "error" term in the LHS of Eq (10)? Isn't the "error" itself an expectation?

2. Although the paper presents generalization bounds, a discussion on the generalization performance of the classifier (VGG-13 for Figure 1) seems to be missing. Only the neural collapse and geometric complexity plots are illustrated but a deeper relationship between the trends and the test performance is lacking. Is there an upper limit of "good" learning rates, and batch sizes above which the geometric complexity can reduce but the test performance can get worse?

3. Similar to the above point, currently it seems like increasing the learning rate, increasing batch size, and increasing the regularization parameter value results in lower neural collapse values. Ideally, neural collapse is applicable during the terminal phase of training (TPT) where the training accuracy is $1$. Does this mean that all the experiments (at least for Figure 1) lead to TPT during training?

nit: Please fix the notation for the norms as it is unclear if $||.||$ represents $||.||_2$ or $||.||_F$.

**Questions:**

1. Since the geometric complexity-based generalization/transfer learning bounds are the upper bounds of existing neural collapse-based bounds, what can be a potentially new implication of your results? since it is common practice to do a hyper-parameter search over learning rates, batch sizes, and regularization parameter values.

2. The paper does not discuss any explicit regularization mechanism that aims to reduce the geometric complexity. What are the potential challenges in designing such a regularization mechanism?

---

> ### Author Rebuttal · Authors · 2024-08-06
>
> Thank you for your comments. We are very happy that you found that the “unified analysis of neural collapse and geometric complexity presents an interesting line of research for the community [...] in terms of studying seemingly disjoint phenomena under a common lens.”  We have addressed your comments to the best of our ability below.
>
> Please consider raising your score if your concerns have been resolved.
>
> We'd also like to respectfully note that your concerns seem to focus around only minor aspects of our work that are easily fixable (i.e., clarification, notation, typos, addition of plots). Could you please help us understand which of your concerns prompted you to a "confident reject” so that we can address it?
>
> > *"There seems to be a mismatch in the log⁡(2/𝛿) term in Proposition 4.2 vs log⁡(1/𝛿) term in Proposition 4.3, which makes the main results inconsistent (yet fixable). Additionally, why do we have an "expectation" over the "error" term in the LHS of Eq (10)? Isn't the "error"  itself an expectation?"*
>
> Thank you for noticing the typo: there should be $\log(2/\delta)$ instead of $\log(1/\delta)$ in Prop. 4.3. As for the second expectation in the LHS of Eq. (10) it is taken over the sample S used to produce the classifier $h_{f,S}$ on line 230  (similarly to  [22, Proposition 5]). We will correct the typo and make the second point clearer in the paper.
>
> > *"Although the paper presents generalization bounds, a discussion on the generalization performance of the classifier (VGG-13 for Figure 1) seems to be missing. Only the neural collapse and geometric complexity plots are illustrated but a deeper relationship between the trends and the test performance is lacking."*
>
> We included plots for the test accuracy in the rebuttal document (see Figure 1, 3rd column). As referenced in our paper, it has already been observed in [22] that lower levels of NC are indicative of higher test accuracy. Similarly, in [17] it has been shown that lower levels of GC also are characteristic of higher test accuracy. We will add a remark to this effect in the paper and include similar plots for our experiments in the appendix.
>
> > *"Is there an upper limit of "good" learning rates, and batch sizes above which the geometric complexity can reduce but the test performance can get worse?"*
>
> Yes, you are correct. As for any form of regularization (implicit or explicit), test performance degrades beyond a certain level of the regularization rate, while the regularized quantity keeps decreasing at the expanse of the original objective.
>
> >*"Similar to the above point, currently it seems like increasing the learning rate, increasing batch size, and increasing the regularization parameter value results in lower neural collapse values. Ideally, neural collapse is applicable during the terminal phase of training (TPT) where the training accuracy is 1. Does this mean that all the experiments (at least for Figure 1) lead to TPT during training?"*
>
> Yes, that is correct and a good clarification. All our experiments lead to training accuracy 1. We included these plots for our main experiments in the attached rebuttal document (see Figure 1, 1st and 2nd columns) and will add similar plots for our additional experiments to the appendix.
>
> > *"Please fix the notation for the norms as it is unclear if $||\cdot||$ represents $||\cdot||_2$ or $||\cdot||_𝐹$."*
>
> Thank you. We will clarify the norm notation in the paper as you suggest.
>
> > *"Since the geometric complexity-based generalization/transfer learning bounds are the upper bounds of existing neural collapse-based bounds, what can be a potentially new implication of your results? since it is common practice to do a hyper-parameter search over learning rates, batch sizes, and regularization parameter values."*
>
> To our knowledge, there are no known theoretical mechanisms which control neural collapse. Our bound in Prop. 4.1 shows that the same mechanisms that control GC also control NC. This is significant because (see [17] from the paper references) there are a number of experimental mechanisms which control the GC and furthermore a strong theoretical justification as to why [17, Thm 5.1]. The current work provides a novel justification that these same mechanisms also control NC. This implication is theoretically motivated via our Prop 4.1 and experimentally validated.
>
> A second new implication that can be derived from our bound in Prop 5.1 is that pretrained networks with lower GC have better transfer learning performance on new tasks. These 2 new implications are indeed the key theoretical findings in our paper. Devising practical schemes based on these theoretical findings is beyond the scope of the current paper, yet opens a new avenue for future work.
>
> > *"The paper does not discuss any explicit regularization mechanism that aims to reduce the geometric complexity. What are the potential challenges in designing such a regularization mechanism."*
>
> This is a good point. While direct regularization by GC computed at the logit layer has been previously studied (for instance in [17]) and found beneficial, direct regularization by GC computed at the embedding layer becomes impractical because of the much higher dimension of the embedding layer, resulting in a very large computational cost.
>
> That being said, explicit regularization by GC computed at the logit layer provides a direct way to control GC computed at the embedding layer. We added this experiment (see rebuttal document Fig. 3) and observed the predicted effect on the embedding GC (rebuttal Fig 3. left and middle) and on neural collapse (rebuttal Fig 3. right). We will add a paragraph to our paper about the challenge of directly regularizing for embedding GC.

---

> > ### Comment · Reviewer_pQCH · 2024-08-09
> >
> > Thank you for the responses and also the additional experimental results.
> >
> > - In my opinion, the authors discussion on explicit GC regularization is key to improve the contribution. Although it is computationally inefficient, it is always helpful to have experiments which validate the claim in this paper. Based on the rebuttal PDF, it seems like NC indeed reduces when explicit GC regularization is employed.
> >
> > - I would request the authors to focus more on this aspect (techniques, limitations, tuning etc of such a regularizer) to further strengthen their paper. Simply showing correlations by tuning common hyper-parameters might not fully justify your work.
> >
> > I have increased the score. Good luck.

---

> > > ### Author Response · Authors · 2024-08-09
> > > **Thank you!**
> > >
> > > We are glad that we were able to address your concerns. And we agree, this in an important (and potentially useful) aspect, despite the inherent computational inefficiencies. Including these experiments has helped to improve the exposition. Thanks again for the suggestion!

---

### Official Review · Reviewer_N73X · 2024-07-13

**Soundness:** 2
**Presentation:** 2
**Contribution:** 2
**Rating:** 5
**Confidence:** 4

**Summary:**

This paper examines the relationship between geometric complexity (GC), neural collapse (NC), and transfer learning performance in deep neural networks. The key contributions are:

* Proposing geometric complexity as a measure that connects the flatness of the loss surface and neural collapse
* Deriving theoretical bounds showing GC controls NC under certain assumptions
* Empirically demonstrating that mechanisms affecting GC (learning rate, batch size, regularization) also influence NC
* Showing lower GC during pre-training leads to better transfer learning, especially for few-shot tasks
* Proving a new generalization bound in terms of GC
* Demonstrating GC can be efficiently estimated on small samples

The authors argue that GC can serve as an informative proxy for transfer learning potential and provide a unifying framework for understanding implicit biases in deep learning. By bridging concepts from optimization geometry, representation learning, and transfer learning, the paper aims to provide deeper insights into the fundamental mechanisms driving the success of modern deep learning techniques.

**Strengths:**

**Originality**: The paper presents a novel perspective connecting several concepts in deep learning theory. Its original contribution is using geometric complexity as a bridge between loss surface geometry and neural collapse. This approach provides a fresh lens through which to view the implicit biases in deep learning, potentially unifying several strands of research. Using GC as a progress metric for transfer learning is innovative and could open up new avenues for improving pre-training techniques.

**Quality**: The empirical evaluation thoroughly examines multiple datasets, architectures, and hyperparameter settings. The authors have made a commendable effort to validate their hypotheses across various conditions, including different network architectures (ResNet, VGG), datasets (CIFAR, MNIST, Fashion-MNIST), and transfer learning scenarios. The ablation studies exploring the impact of learning rate, batch size, and regularization on GC and NC provide valuable insights into the dynamics of deep network training. While limited in scope, the theoretical results give some formal grounding for the empirical observations and offer a starting point for further analytical work in this area.

**Clarity**: The paper is well-structured and communicates the main ideas. The authors have presented a complex set of ideas in a logical flow, starting from the theoretical foundations and moving to empirical validation. The figures illustrate key results, particularly the relationships between GC, NC, and transfer performance. The use of color-coding and consistent formatting across figures aids in interpretation. The pseudo-code and detailed experimental setup descriptions in the appendix are beneficial for reproducibility.

**Significance**: If the claims hold up, this work could provide valuable insights into the mechanisms behind transfer learning and self-supervised learning. The proposed geometric complexity measure may help analyze and improve deep learning models. The potential impact extends beyond just theoretical understanding. If GC can serve as a reliable proxy for transfer learning potential, it could guide the development of more effective pre-training strategies and architectures. Furthermore, the connection between GC and NC could help explain why certain training practices (like large batch training or specific learning rate schedules) are effective, potentially leading to more principled approaches to hyperparameter tuning.

**Weaknesses:**

**Limited theoretical foundation**: While the paper provides some theoretical results, the assumptions required (e.g., Poincaré inequality) are quite strong and may not hold in practice for real datasets and architectures. The connection between the linear model analysis and deep neural networks is not strongly justified. This is a significant limitation, as it's unclear how much of the theory translates to practical scenarios. The authors could strengthen this aspect by providing empirical evidence that these assumptions hold in realistic settings or deriving weaker results under more general conditions. Additionally, the paper could benefit from a more thorough discussion of the implications and limitations of these theoretical results.

**Experimental limitations**: The empirical evaluation focuses on image classification tasks and relatively small datasets. The generalizability of the results to larger datasets, other domains (e.g., NLP), or more complex architectures is unclear. This narrow focus raises questions about the broader applicability of the findings. For instance, it's unclear whether the relationships between GC, NC, and transfer performance would hold for large-scale pre-training scenarios like those used in modern language models. The paper would be significantly strengthened by including experiments in a broader range of tasks and scales or at least by providing a more detailed discussion of the potential challenges in scaling up the approach.

**Lack of comparison to related metrics**: The paper does not thoroughly compare geometric complexity to other related complexity measures or generalization bounds from the literature. This makes it difficult to assess the relative merits of GC. There's a rich body of work on complexity measures for neural networks, including various forms of capacity measures, stability-based bounds, and PAC-Bayesian approaches. A comprehensive comparison would help situate GC within this broader context and clarify its unique contributions. Such a comparison could also help identify scenarios where GC might be particularly advantageous or potentially suboptimal compared to existing approaches.

**Causality vs correlation**: While the paper shows correlations between GC, NC, and transfer performance, it does not conclusively demonstrate a causal relationship. Alternative explanations for the observed trends have not been thoroughly explored. This is a critical limitation, as it leaves the possibility that GC is merely a proxy for some other underlying factor driving NC and transfer performance. The paper would be strengthened by a more rigorous causal analysis, perhaps through interventional studies or by controlling for potential confounding variables. Additionally, discussing potential mechanisms by which GC might causally influence NC and transfer learning would add depth to the analysis.

**Questions:**

1. How does geometric complexity compare to other measures of model/representation complexity like Fisher information, gradient norm, etc.? A more thorough comparison would strengthen the paper.
2. The theoretical results rely on strong assumptions like the Poincaré inequality. Can you provide empirical evidence that these assumptions hold for real datasets/models? Or derive results under weaker assumptions?
3. The paper shows correlations between GC, NC, and transfer performance but does not conclusively prove causation. Have you considered alternative explanations or confounding factors?
4. How computationally expensive is it to measure GC during training? Is it feasible to use it as a regularization term or early stopping criterion in practice?
5. The generalization bound in terms of GC is interesting, but how tight is it empirically? How does it compare to other generalization bounds in the literature?
6. Have you explored using insights about GC to guide architecture design or improve training algorithms? Demonstrating practical benefits would strengthen the paper.

**Limitations:**

The authors acknowledge some limitations, like the focus on image classification tasks and relatively small datasets. However, they could go further in discussing potential shortcomings:

* The strong assumptions required for the theoretical results may limit their applicability.
* The computational cost of measuring GC is not thoroughly addressed.
* Potential negative consequences of optimizing for low GC are not explored.

---

> ### Author Rebuttal · Authors · 2024-08-06
>
> Thanks for your thoughtful review. We are very happy that you found our approach to be a “novel perspective” as well as an “original contribution” and that it “could open up new avenues for improving pre-training techniques” and “provide valuable insights into the mechanisms behind transfer learning and self-supervised learning”. We are also glad that you found the “empirical evaluation” to be “thoroughly” done and that the paper was “well-structured” with “ideas communicated clearly”.
>
> Below are our answers to your concerns. They have been very helpful in further strengthening the paper.
>
> We ask that you please consider raising your score if your concerns have been resolved.
>
> >*"The connection between the linear models and DNNs is not justified"*
>
> There is no mention of linear models in our paper. Can you please reference the paragraph creating confusion so that we can clarify?
>
> >*"Limited theoretical foundation: [..] the assumptions required (e.g., Poincaré inequality) are quite strong [..] the authors could strengthen this by providing evidence that these assumptions hold in realistic settings"*
>
> This is a good point but we respectfully disagree with the idea that the Poincare Inequality (PI)  is an overly strong or unrealistic assumption. We agree our paper would be strengthened by clarifying this point. Here, we include a short summary of key facts and can provide a more detailed exposition in the Appendix.
>
> Note that the Gaussian distribution; mixtures of Gaussians with equal (or different) variances; mixtures of Gaussians and sub-Gaussians; mixtures of uniform and Gaussian measures; and any log-concave probability measure all satisfy a PI ([1, 2] below). The same is true for most distributions of importance in probability and statistics ([3] below).
>
> The PI has also been assumed to hold for real life image datasets, where it has been used to help improve GAN convergence ([4] below). It is also a key assumption to understand the role of over-parameterization in generalization as happens for large NNs ([5] below).
>
> On the contrary, non-PI distributions are considered pathological; they can be constructed for instance by artificially concatenating distributions with fully disjoint support. The only restrictive assumption for the model is that it's differentiable wrt the input, which is a widely assumed property in the literature.
>
> Furthermore, we examine various real-life datasets like CIFAR10, CIFAR100, and ImageNet as well as common architectures like VGG and ResNet.
>
> [1] Bakry et al. A simple proof of the Poincaré inequality for a large class of probability measures, '08
>
> [2] Schlichting, Poincaré and log-Sobolev inequalities for mixtures, '19
>
> [3] Pillaud-Vivien et al, Statistical Estimation of the Poincaré constant and Applications..., AISTATS '20
>
> [4] Khrulkov et al, Functional Space Analysis of Local GAN Convergence, ICML '21
>
> [5] Bubeck et al, A universal law of robustness via isoperimetry, NeurIPS '21
>
> >*"Experimental limitations: [..] The generalizability of the results to other domains (eg, LLMs) is unclear [..]"*
>
> This is good point. Thanks for raising it. We will definitely clarify this aspect in our paper. There is a limitation of our work when it comes to extending to LLMs. However, this limitation is not specific to our work per-se but it is a general limitation of the application of NC to LLMs (see Wu & Papyan, Linguistic Collapse: Neural Collapse in (Large) Language Models, ‘24 which outlines limitations).
>
> The main problem is that for LLMs the embedding dimension is lower in general than the number of classes (vocabulary size), making the NC simplex impossible to exist. Extending the notion of NC to LLMs is an open question beyond our scope.
>
> >*"How does GC compare to other measures [..]"*
>
> This is indeed interesting. However, this comparison has been done already; e.g. [17] introduces GC and thoroughly relates it to the gradient norm & [41] compares with other complexity measures. The relation with the Fisher information and the gradient norm has also been studied (Jastrzebski et al., Catastrophic Fisher Explosion, ICML '21). These works indicate that the GC as a measure in function space is a similar notion of complexity to the gradient norm or the Fisher information in parameter space.
>
> We are happy to add an Appendix section on this.
>
> >*"The paper shows correlations with GC [..] but not [..] causation"*
>
> We added an experiment (rebuttal:Fig. 3) that directly controls GC by explicitly regularizing it, mitigating confounding factors. We observe the same predictions: lower GC produces lower NC. Note, here we regularized w.r.t. the GC computed at the logit layer rather than the embedding GC. This is because the high dimensionality of the embedding layer makes taking the gradient of the embedding GC very prohibitively expensive.
>
> >*"How computationally expensive is it to measure GC [..]? Is it feasible to use as a regularization term?"*
>
> We have not yet tried to use GC as an early stopping criterion but this could be a good idea. Measuring GC during training is relatively cheap, easy, and robust, especially if one samples it as explained in Section 4.2, see also Fig. 2. We have explored using GC as a regularization term (see above & rebuttal:Fig. 3) as have others, eg [17].
>
> >*"The generalization bound in terms of GC is interesting, but how tight is it empirically?"*
>
> We added an experiment (rebuttal:Fig. 2) where we plotted the LHS and RHS (excepting the Lipschitz term) of this bound Eq. 10 in Prop 4.3, which we will add to our paper.
> On that plot the bound is not vacuous and quite tight! A full comparison with other generalization bounds is beyond our scope. Our main goals are showing that 1) the mechanisms that control GC also control NC and 2) solutions with lower GC are more performant for transfer learning.
>
> >*"Have you explored using GC insights to guide models or algorithms?"*
>
> This is a very interesting but beyond the scope of the current paper.

---

> > ### Author Response · Authors · 2024-08-12
> >
> > Thank you again for your review. We hope that the concerns in your original review were sufficiently addressed in our response. In particular, the justification of  the Poincare inequality assumption for data distributions, the demonstrated empirical tightness of the Generalization bound (our Propn 4.3), and the explicit GC regularization experiments exploring causality of these phenomena.
> >
> > We also hope we addressed the concerns regarding experimental limitations particularly with respect to the applicability to modern language models and LLMs. As the discussion period ends, we wanted to see if there is anything that needs additional clarification. Thank you.

---

### Official Review · Reviewer_5tN3 · 2024-07-21

**Soundness:** 3
**Presentation:** 3
**Contribution:** 3
**Rating:** 6
**Confidence:** 3

**Summary:**

The paper explores the relationship between neural collapse (NC) and geometric complexity (GC). It presents both theoretical and empirical evidence showing that geometric complexity is a robust metric across various variables. By substituting the NC metric with GC, the paper introduces a generalization bound based on geometric complexity. Furthermore, it highlights the significance of GC in transfer learning, demonstrating that by controlling the pre-training GC, downstream NC can also be controlled, leading to improved transfer learning results.

**Strengths:**

* The paper is overall well-written and easy to follow, the interpretation of each theoretical results is clear and intuitive.
* The relationship between NC and GC is quite interesting.
* Most of the empirical results align well with the theoretical part, demonstrating the validity of the results.

**Weaknesses:**

Some empirical results can be better aligned with the theoretical counterpart and additional experiments could be added to better support the claims. For example:
* In proposition 4.1, NC is upper bounded by geometric collapse up to some constant scaling. What's the range of this constant $c$, does it purely depend on $Q$? If yes, it would be better to directly plot NC and RHS in (6) in the same plot to demonstrate the validity of the bound; and if no, it would be interesting to show how different settings examined in Figure 1 affect $c$.
* The empirical evidence presented in Figure 2 clearly demonstrates the robustness of geometric complexity. However, I have some concerns regarding the Lipschitz notion. Neural networks are known to be not so Lipschitz, and making them more Lipschitz is an active area of research aimed primarily at improving robustness. Is Proposition 4.2 numerically verifiable by calculating the Lipschitz constant of the tested neural networks?

**Questions:**

* In equation (7), does the distance $d$ represent the norm distance? This should be stated clearly.
* Both the paper and the prior work (Galanti et al, 2022) rely on the data assumption where the source and transfer datasets come from the same distribution $Q$. However, in practical settings, source and transfer datasets typically have domain gaps which can potentially violate the data assumption. Would the results in the paper still hold in this case? For example, pre-training on ImageNet/Cifar-100 while fine-tuning on DTD?

**Limitations:**

Yes

---

> ### Author Rebuttal · Authors · 2024-08-06
>
> Thanks for your review. We are very happy that you found the work “well-written” and “easy to follow”, making the interpretation of the theoretical results “clear” and “intuitive”. We are also glad that you found the relation between “GC” and “NC” to be “quite interesting” and that the “empirical results align well with the theoretical part”.
>
> Below you’ll find below our best effort to answer your questions.
>
> Please consider raising your score if you feel your concerns have been resolved.
>
>
> >*"In proposition 4.1, NC is upper bounded by geometric collapse up to some constant scaling. What's the range of this constant c, does it purely depend on Q.  If yes, it would be better to directly plot NC and RHS in (6) in the same plot to demonstrate the validity of the bound; and if no, it would be interesting to show how different settings examined in Figure 1 affect c."*
>
> Yes, the Poincare constant depends exclusively on the distribution Q. Stacking the plots in Figure 1 as you are suggesting is a great idea, which we will add in our paper. On these plots, by comparing the magnitude RHS and LHS of (6), we can roughly see that the c ~ 1000 (lower bound) for CIFAR-10 in our setting. The Poincare constant can take very high values depending on the spread of the distribution. For instance, for a standard normal distribution the Poincare constant is 1 while for a multivariate normal distribution the Poincare constant equals the largest eigenvalue of its covariance matrix which can be any positive real number. Intuitively speaking, the Poincaré constant will increase if the space is stretched in any direction and decrease if it is compressed in any direction.
>
> >*"The empirical evidence presented in Figure 2 clearly demonstrates the robustness of geometric complexity. However, I have some concerns regarding the Lipschitz notion. Neural networks are known to be not so Lipschitz, and making them more Lipschitz is an active area of research aimed primarily at improving robustness. Is Proposition 4.2 numerically verifiable by calculating the Lipschitz constant of the tested neural networks?"*
>
> The numerical experiment in Figure 2 (plot 1) shows that in practice the empirical GC estimate may be approaching the true value much faster than the bound specifies. Namely after 10 samples only, we already obtain a value very close to what we obtain with 50 times more samples, which is much better than a 1/sqrt(#sample) dependence, even ignoring the Lipschitz constant. This makes us believe that a much tighter bound here should be possible. We agree with you that this bound may be a looser bound than reality.
>
> >*"In equation (7), does the distance d represent the norm distance? This should be stated clearly."*
>
> Thank you for pointing this out. We will clarify that point in the paper. “d_ij” represents the euclidean distance between the mean for class i and the mean for class j.
>
> >*"Both the paper and the prior work (Galanti et al, 2022) rely on the data assumption where the source and transfer datasets come from the same distribution Q. However, in practical settings, source and transfer datasets typically have domain gaps which can potentially violate the data assumption. Would the results in the paper still hold in this case? For example, pre-training on ImageNet/Cifar-100 while fine-tuning on DTD?"*
>
> This is a very good point. In theory any source and target distribution can always be concatenated together mathematically into a single larger distribution. However, as you are pointing out, issues may arise when “gaps” are created by this process. Namely, when the support of the source and target distribution do not overlap, this process results in an overall distribution with non-connected support. This particular configuration is a setting ripe for violations of the Poincare inequality even if the source and target distribution both satisfy it.
>
> If this happens, then the range of validity of our bounds as well as our predictions concerning the impact of the learning rate, batchsize, regularization on the neural collapse and transfer performance may not hold.
>
> However, instead of seeing this as a limitation, we interpret it as a feature that may help us understand what compatibility conditions are necessary for a model pre-trained on a source distribution to transfer well to a target distribution. Namely, in line 286, we give compatibility conditions for the transfer bound in Prop'n 5.1 to be meaningful. This condition in particular implies that no such gaps are created by concatenating the source and target distributions.

---

> > ### Author Response · Authors · 2024-08-12
> >
> > Thank you again for your review. We hope that the concerns in your original review were sufficiently addressed in our response, particularly, the application of  the Poincare inequality and the role of the constant c in our Propn 4.1. As the discussion period ends, we wanted to see if there is anything that needs additional clarification. Thank you.

---

> > ### Comment · Reviewer_5tN3 · 2024-08-13
> > **Response to rebuttal**
> >
> > Thanks to the authors for the detailed response.
> >
> > Given that the authors said "c ~ 1000 (lower bound) for CIFAR-10 in our setting", would this make the bounds in Proposition 4.1 and 4.3 vacuous?

---

> > > ### Author Response · Authors · 2024-08-13
> > >
> > > This is a good question but indeed, taking the value c near 1000 does not make the bounds in Proposition 4.1 and Proposition 4.3 vacuous. Figure 1 in our paper plots the Geometric Collapse (i.e. the RHS of the inequality in Propn 4.1 excepting this constant c) in the middle column and the Neural Collapse (i.e. the LHS of the inequality of Propn 4.1) in the right column. From these plots, we see that by comparing the magnitude of the Neural Collapse and the Geometric Collapse that indeed c ~ 1000 for CIFAR-10 in our setting.
> > >
> > > We also explored directly the tightness of the bound derived in Proposition 4.3 during the rebuttal. Please see the 1-page pdf which we attached to our rebuttal response. Figure 2 of that rebuttal pdf shows empirically the tightness of the bound in Propn 4.3. For that experiment we trained a VGG-13 model on CIFAR-10 with 5 random seeds. The plot shows the LHS of the inequality in Propn 4.3 (i.e. the nearest mean classifier error on the test set ) in blue and the RHS of the inequality in Propn 4.3 (excepting the Lipschitz term which we expect to be negligible due to Fig 2 and (8) of our paper) in orange.
> > >
> > > Note that here we take $c = 1000$, $p = 1024$ (which denotes the embedding dimension of the feature map) and $m_c = 5,000$ (which denotes the number of samples per class). Indeed, the bound is not vacuous and in fact appears quite tight!
> > >
> > > Thanks again for your careful and thoughtful read. I hope we've been able to successfully address your concerns.

---

> > > > ### Comment · Reviewer_5tN3 · 2024-08-13
> > > >
> > > > Thanks, that has addressed my concern.
> > > >
> > > > Raising the score now.

---

> > > > > ### Author Response · Authors · 2024-08-13
> > > > >
> > > > > Thanks again and thank you for adjusting your score. We are very glad that we were able to address your concern.

---

### Author Rebuttal · Authors · 2024-08-06

We thank the reviewers for your thoughtful and thorough reviews. We are grateful that you found the paper "well-written" and "well-structured" and found our "novel perspective" to be an "original contribution" which you think “could open up new avenues for improving pre-training techniques” and “provide valuable insights into the mechanisms behind transfer learning and self-supervised learning”. This is great to hear and much appreciated! Furthermore, your thoughtful comments and review have helped us to substantially improve and strengthen the current work.

The overall goal of this paper is to connect two previously unconnected areas of research in machine learning and leverage that insight to better understand their role in successful transfer learning. We achieve this goal in two ways:

First, we clearly demonstrate the relationship between neural network geometric complexity and embedding neural-collapse (Section 4). We show through theory and verify empirically that mechanisms which regularize the GC in turn put measurable pressure on the neural collapse (Section 4.1: Prop 4.3 and Fig. 1).

A second goal is to show that pre-trained networks with lower GC promote lower NC values on new unseen target classes which enable improved transfer accuracy during fine-tuning (Section 5: Prop 5.1 and Fig. 3).

We tried our very best below to satisfy your requests and answer your questions within the time frame imparted.

**Please find attached a 1-page document containing figures referenced in our responses to individual reviewers comments.**

Please consider raising your scores in consequence if your requests have been met and questions answered.

---

### Decision · Program_Chairs · 2024-09-25

**Decision:**

Accept (poster)

**Comment:**

This paper studies the relation between metrics of neural collapse (NC) and geometric complexity (GC).
They prove that NC can be bounded in terms of GC under certain assumptions, and they give a generalization bound depending on GC.
They also empirically demonstrate that GC and NC tend to be correlated under common training interventions (learning rate, etc).
Overall, this paper presents a promising case for using GC in understanding neural collapse.

The reviewer scores were borderline. Reviewers agreed that the relation between GC and NC was interesting and novel. The main concerns were about the lack of comprehensive experimental support, and lack of sufficient discussion of limitations.

I recommend acceptance, since reviewers found the ideas interesting and found no technical flaws.
I strongly suggest the authors incorporate reviewer feedback in their camera-ready. Specifically, please add explicit discussion of both the theoretical and the experimental limitations. Be especially careful in making causal claims without specifying your causal graph. Overall, adding such discussion will clarify the scope of your work, and thus make the paper stronger.